# Where to start? Analyzing the potential value of intermediate models

**Leshem Choshen**[*†], **Elad Venezian**[*], **Shachar Don-Yehiya**[‡], **Noam Slonim, Yoav Katz**

IBM Research, †MIT, ‡The Hebrew University
{leshem.choshen@, eladv@il.,shachar.don-yehiya@ noams@il., katz@il.}ibm.com

## Abstract

Previous studies observed that finetuned models may be better base models than the vanilla pretrained model. Such a model, finetuned on some source dataset, may provide a better starting point for a new finetuning process on a desired target dataset. Here, we perform a systematic analysis of this *intertraining* scheme, over a wide range of English classification tasks. Surprisingly, our analysis suggests that the potential intertraining gain can be analyzed *independently* for the target dataset under consideration, and for a base model being considered as a starting point. Hence, a performant model is generally strong, even if its training data was not aligned with the target dataset. Furthermore, we leverage our analysis to propose a practical and efficient approach to determine if and how to select a base model in real-world settings. Last, we release an updating ranking of best models in the HuggingFace hub per architecture.[1]

## 1 Introduction

Finetuning pretrained models (Devlin et al., 2019), is currently the standard and best approach for adjusting such models to perform a downstream task (Chen et al., 2022). The resulting finetuned models are typically used for inferring the labels of new examples that are reminiscent of the data used for finetuning. However, it was shown (Phang et al., 2018a) that finetuned models, trained on some *source dataset*, may represent better *base models*, namely a better starting point for a new finetuning process on a desired *target dataset*. This scheme, often referred to as *intertraining*, is the focus of the present work.

Given a target dataset, one may wonder what could be the intertraining gain, to determine whether it is worthwhile spending resources on selecting a base model. Assuming the potential gain is high, the following natural question is which base models are most promising, out of countless options available through hubs HuggingFace (e.g. Wolf et al., 2020). We propose pragmatic methods to answer both questions, supported by extensive experiments.

We begin with two observations: (i) some target datasets are *intertraining-sensitive*, i.e., have the potential to gain significantly from intertraining, while others are not, and are typically indifferent to the base model selection. Furthermore, revealing this property of the target dataset can be done efficiently, by examining the gains obtained when using a *single* representative base model as a starting point; (ii) some base models are of high *quality*, i.e. finetuning on them provides consistent improvements on target datasets, but most base models are inferior and degrade performance. Furthermore, ranking base models by quality can be done on one target task – and efficiently, via *linear probing*, namely training only the base model classification head, over a single representative dataset.

Thus, we argue that a preferable base model can be **selected independently** of the target dataset. This is in contrast to the common perception (c.f. §7) that the alignment of the target dataset and the source dataset – used to generate the base model – is a major factor in determining intertraining success. We substantiate our observation of independence by conducting experiments on a comprehensive set of target datasets and base models, comprising models obtained under controlled conditions as well as models from HuggingFace. In addition to these findings, we analyze attributes of the source and target datasets that affect gains (§6).

As some models are just better, not due to the choice of a current dataset, it makes sense to **rank the models once** and pick the best ones. But even

---

*These authors contributed equally to this work.
[1] https://ibm.github.io/model-recycling/

ranking a thousand models is costly. In §8, we rely on our analysis to propose a practical approach to efficiently select models in a real-world setting. Moreover, instead of expecting others to rank the models, we share an updating site site featuring the best models currently found. So far, we tested over 2.5K models.

## 2 Preliminaries

In this paper, we use the following terminology. A *dataset* is a set of examples and labels. Our goal is to maximize accuracy on the test of the *target dataset*, "target" for short. We discuss the difference between domain, task, and dataset in App. A.

A *pretrained (PT) model* is a self-supervised model, e.g., RoBERTa (Liu et al., 2019). A *finetuned model* is a PT model that was further trained over some *source dataset*, denoted henceforth as "source". We assume access to many such models, e.g., through HuggingFace. A *base model* can be either a PT model or a finetuned model. When finetuning over the target train data, one can start from any base model. *Intertraining* refers to starting from a finetuned model as a base model, and in this case, we refer to this base model as an *intermediate model*. We denote by $S_m^t$, the accuracy score obtained over the target test set, $t$, after finetuning some base model $m$ over the target train set. The intertraining *gain* of model $m$ w.r.t. using the PT model, is thus defined via $gain\,(m,t) = s_m^t - s_{PT}^t$. Note that the gain may be negative. Given a set of intermediate models, $M = m_1 \ldots m_n$, the intertraining *max-gain* is defined as $\max_{m \in M}\left(s_m^t - s_{PT}^t\right)$. Thus, theoretically, max-gain is achieved by finetuning all the available intermediate models and picking the one best performing on the target test set. To avoid overfitting and reduce costs, our goal is to find an intermediate model with a gain that is as close as possible to the max-gain, without explicitly finetuning all the intermediate models.

## 3 Experimental Setup

Our experimental setup is described next. The parameters for reproducibility are detailed in App. B.

### 3.1 Dataset Groups

Our experiments are based on 3 groups of English text classification datasets described next (App. C).

We focus on text classification for ease of evaluation, but assume the tasks are diverse enough for our conclusions to extend to other settings.

**General.** Containing GLUE (Wang et al., 2018) and SuperGLUE classification datasets (Wang et al., 2019a), excluding test only and regression datasets. The datasets cover a wide range of tasks, from sentiment analysis through linguistic acceptability to natural language inference. It is the most commonly used benchmark in related work (§7).

**NLI.** Containing Natural Language Inference and Entailment tasks. Datasets of this group all share the same task. There is some overlap between NLI and General; in Fig. 1 and mean comparison we subtract the overlapping datasets from General.

**Twitter.** Containing 6 Twitter datasets collected by TweetEval (Barbieri et al., 2020). The tasks range from irony detection to emoji prediction. Datasets of this group all share the same domain.

### 3.2 Models

Unless stated otherwise, our PT model of choice is RoBERTA (Liu et al., 2019). We acquire intermediate models in two ways:

**In-house.** Obtained by finetuning the PT model over General, NLI, and Twitter dataset groups as the source datasets, with 5 seeds. Since we control the datasets and have knowledge about their features, this enables us to find relations between the dataset properties and the intermediate models generated from these datasets.

**Off-the-shelf.** 66 RoBERTa models downloaded from HuggingFace (see App. §E for more details). Since these models do not carry information excluding their names, this set allows us to validate our claims on a "real-world" model distribution.

### 3.3 Models/Targets experiments

We test many intermediate models on various target datasets. We finetune each intermediate model and the PT on the target train, and report the intertraining gain over the target test. In the *In-house models/targets experiment*, all 22 datasets from the General, NLI, and Twitter groups act as both source and target and gains are average of 5 seeds. In the

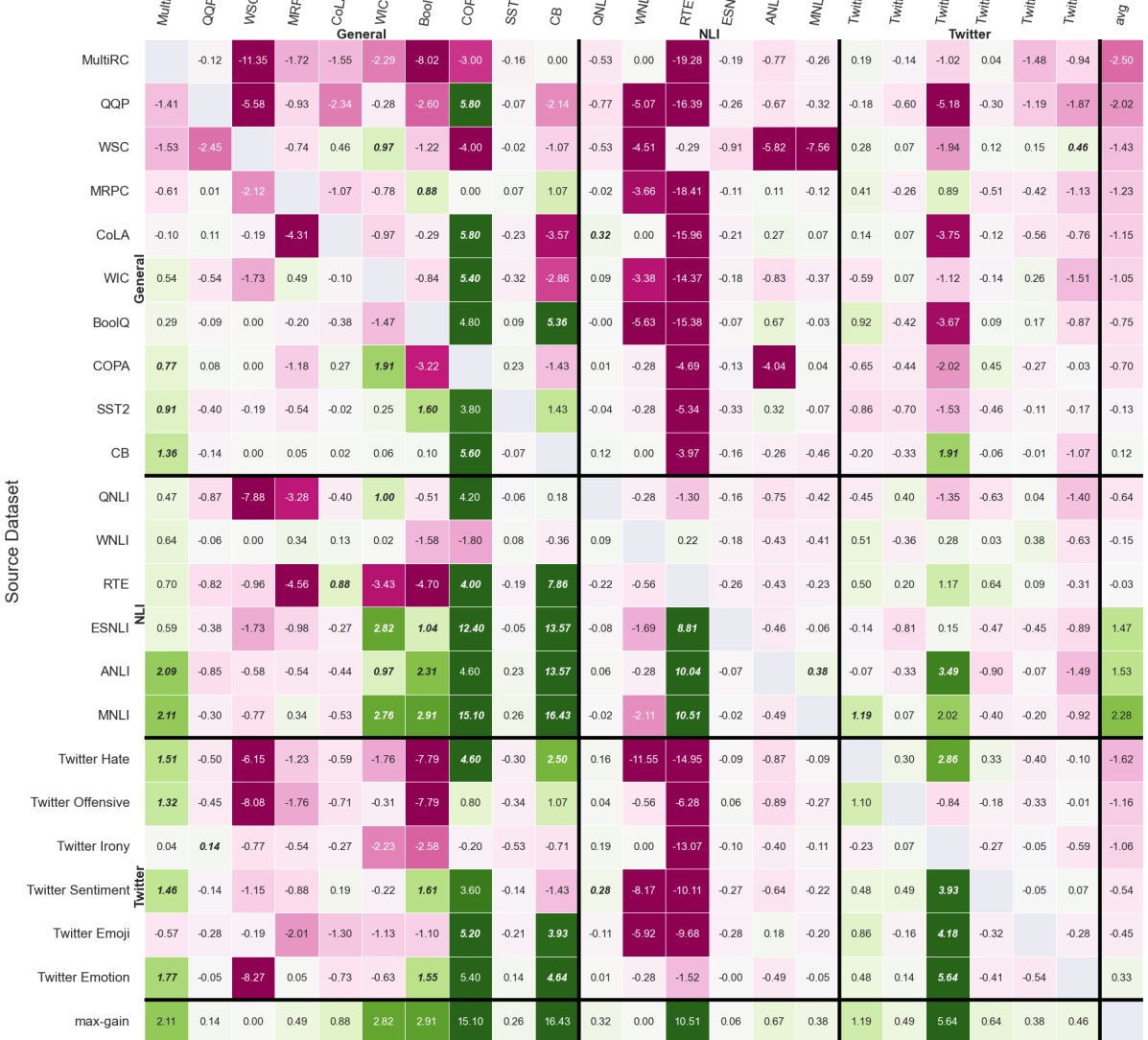

Figure 1: Results of in-house models/targets experiment. Columns correspond to target datasets and Rows correspond to intermediate models generated based on same datasets as source. The 22 datasets come from the General, NLI and Twitter groups. Each value indicates intertraining gain w.r.t. the PT model, averaged over 5 seeds. Sorted by group and source average gain (bottom row). Positive significant cells (>2 STD) are italicized.

*Off-the-shelf models/targets experiment*, we download the 66 source models from Huggingface and test on the 14 General datasets as targets.

## 4 Results

Most models are worse than PT and about 1 in 6 are better, providing positive intertraining gain. The in-house models/targets results are depicted in Fig. 1 and STDs and reference results in App. §D. App. §E reports results with off-the-shelf RoBERTa and T5 intermediate models.

The rows and columns in Fig. 1 are similarly ordered – first the General datasets, then the NLI datasets, and last the Twitter datasets. Loosely speaking, we do not recognize an approximate green block structure across the diagonal; namely, we do not observe clear intertraining gain for similar tasks (NLI); nor for similar domains (Twitter). However, some columns and some rows depict higher intertraining gains, while for others, the impact is minor. Taken together, these observations suggest little dependence between the source used to generate the intermediate model and the perfor-

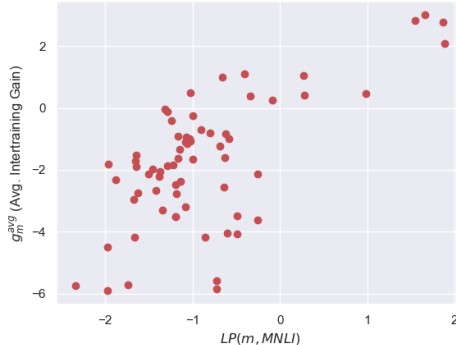

Figure 2: Linear probing MNLI (x) is enough to predict finetuning gains (y) averaged over 14 General datasets. Each point corresponds to one off-the-shelf base model.

mance over the target. This is in contrast to the common assumption (§7) that the source and target need to be similar for intertraining to work. Next, we delve deeper into these observations.

### 4.1 Target Sensitivity to Intertraining

Considering Fig. 1 columns, we notice that for some target datasets (e.g., ESNLI) intertraining makes little difference, while for others (e.g., COPA) the impact is quite significant. We argue that this target property can be predicted via an efficient and straightforward method. Specifically, the gains of one strong intermediate model should resemble the max-gains of a group of models. Indeed, MNLI highly correlates both with the max-gain of in-house models tested on the 22 targets in Fig. 1 (Spearman: 0.89, Pearson 0.99) and off-the-shelf models tested on the 14 General targets (Spearman: 0.90, Pearson: 0.94, $p < 0.01$ for all). The replication on off-the-shelf models shows that this is a general result and not a reflection of MNLI being top of the in-house group. Overall, we find sensitivity is a characteristic of the target dataset separated from the source factor.

### 4.2 Ranking Intermediate Models

Considering Fig. 1 rows, we notice that some intermediate models – e.g., MNLI – provide relatively high gains for many targets. We observe that this is a general phenomenon – stronger models are typically stronger for many datasets.

Identifying such models in advance could be practically valuable, since for a new target, one would consider only the highly ranked intermedi-

ate models (see §8). In the following, we propose a simple yet efficient method to obtain such a static ranking - which is made once, without accounting for the target. A more comprehensive ranking alternative is described in App. §F.

Given an intermediate model $m$, we train a linear probe (LP) – i.e., train only the classification head – over MNLI, and consider the gain, denoted $LP(m, MNLI)$[2]. Evidently, this gain is a good proxy for the quality of $m$. Specifically, let $g_m^{avg}$ be the average gain of $m$ over a set of target datasets. As depicted in Fig. 2, we observe that $LP(m, MNLI)$ and $g_m^{avg}$ are highly correlated in the in-house models/targets experiment (Spearman: 0.46, Pearson: 0.78, $p < 0.01$) and the off-the-shelf models/targets experiment (Spearman: 0.51, Pearson: 0.66, $p < 0.01$). In other words, if available intermediate models are ranked by LP on one dataset $LP(m, MNLI)$, highly ranked models represent the most promising starting points on average. The high correlation means this connection holds not only for the top-ranked models, but throughout. Moreover, the replication on off-the-shelf models shows this is robust not only across targets but across sources.

To further support this claim, we use this ranking to find the most promising intermediate models. For each target $t$, we consider the gain obtained by the top-ranked model and the max-gain obtained by one of the three top-ranked models, denoted $g_{(1)}^t$ and $g_{(3)}^t$, respectively. In comparison, we consider the max-gain obtained while considering *all* available models, denoted $g_{(max)}^t$. We further denote $loss_1^t \equiv g_{(max)}^t - g_{(1)}^t$ and $loss_3^t \equiv g_{max}^t - g_{(3)}^t$. In other words, $loss_1^t$ represents the potential gain loss when using the top statically ranked model versus using the best available model for the target under consideration. Similarly, $loss_3^t$ represents the lost gain when using the best model out of the top 3 ranked models versus using the best available model. Note, that even in this latter case, finding the best model involves finetuning only 3 models over the target train data, which is far less demanding compared to finetuning all available models.

In Table 1, we report statistics for $loss_1^t$ and $loss_3^t$ over the in-house and off-the-shelf models/targets experiments. Although the ranking is target-independent, the top 3 ranked models typ-

---
[2]MNLI is not unique, many datasets showed promising results in initial trials.

ically provide most of the potential intertraining gain. For example, instead of checking all 66 available models, using this simple strategy of checking the top 3 ranked models, each of the 14 targets lost at most 1.62 points.

| Models | @Top | Avg. | Max | # targets s.t. $loss_n^t > 1$ |
|---|---|---|---|---|
| In-house | $loss_1^t$ | 0.37 | 2.11 | 3/22 |
| | $loss_3^t$ | 0.2 | 1.15 | 1/22 |
| Off-the-shelf | $loss_1^t$ | 2.33 | 12.0 | 8/14 |
| | $loss_3^t$ | 0.34 | 1.62 | 2/14 |

Table 1: Lost Gain ($loss_n$) is small when choosing the $n$ top-ranked models. Columns present the aggregation across target datasets of the lost gain: average, max and the number of targets that lose at least one point. Rows present either in-house (22 models and 22 targets) or off-the-shelf (66 models and 14 targets) experiments.

## 5   Source and Target Interaction Analysis

Next, we analyse the interaction between the source dataset and the target dataset. Obviously, such interaction may impact the gain. On one extreme, since finetuning on the same data twice does not improve performance, intertraining is not valuable when the source and target data are identical. On another extreme, consider partitions of the same data. Obviously, training over half the data as the source would be beneficial for the other half, the target. Thus, we do not question that interaction may exist, but rather investigate how common and strong it is.

**Interaction between dataset groups.**   Our General dataset group consists of diverse datasets, while the other groups share a domain (Twitter) or a task (NLI). We analyze whether datasets that share a domain or task with the target represent better source datasets than others.

Table 2 depicts the average gain of each source group vs. each target group. Comparing targets (table columns), we find the models trained on similar tasks, as a group, have a distinct behavior (ANOVA $p < 0.01$). On average, using NLI intermediate models on NLI targets, yields a gain of 0.63, compared to a strong *negative* gain when using intermediate models from other groups. Similarly, there is a same-group gain of 0.5 on Twitter.

Comparing sources (table rows), while NLI is best improved by NLI models, NLI models im-

| | General | NLI | Twitter |
|---|---|---|---|
| General | -0.37 | -2.68 | -0.54 |
| NLI | **1.26** | **0.63** | -0.03 |
| Twitter | -0.4 | -2.39 | **0.53** |

Table 2: Intermediate models trained on sources from the same domain (Twitter) or task (NLI) as the target, yield greater gain. Numbers represent the average gain of intermediate models of a source group (rows) on a given target group (columns) .

prove General datasets even more than NLI ones. Possibly, NLIs are just good intermediate models. Twitter models, however, do seem to improve Twitter targets more (ANOVA $p < 0.01$), by 1 point. Hence, it seems the effects are mixed.

In summary, as a group, a similar source tends to yield greater gains than an unrelated source. However, in the rest of this section, we find this effect is of secondary importance to predicting model gains. A similar source may be more beneficial than a random one, but a well chosen source produces larger benefits regardless of similarity.

**Symmetry as a similarity bound.**   We consider dataset similarity from another perspective. Similarity is a symmetric relation. Hence, if source-target similarity was a main reason for gains, we would expect that when the source $A$ helps the target $B$, the source $B$ would help $A$ as well. We assess the symmetry of the in-house models/targets experiment. We find that gains are far from symmetric (details in App. §G). Thus, (symmetric) similarity seems like a weak explanation of which source data would help which target.[3]

**Regression.**   As additional support for the relatively small impact of the source-target interaction, we show that the interaction is hardly needed to predict the scores. Specifically, a simple regressor can approximate the gain without considering such interactions. The regression fits 3 sets of coefficients. For each target two coefficients $t_i, t_i'$ – which one may think of as capturing average gain and sensitivity to inter-training, per target; and for each base model $b_j$ – which one may think of as capturing average inter-training gain, per base model. We then define $\widehat{gain}(base_i, target_j) = (b_i + t_j) t_j'$ where $i$ and $j$ are the base model and target indices, respectively. Note that by construction, this regressor

---
[3]Even if similarity was a strong indicator we would not expect a full symmetry, as other factors such as sizes do matter.

has $2\dot{N}+n$ parameters, while trying to model $N\dot{n}$ interactions; thus, it does not have enough degrees of freedom to explicitly model all source/target interactions. Still, it obtains satisfactory performance, as shown next. As a baseline, we fit the same regressor after randomly shuffling all the gains in the experiment. This shuffling ensures no information comes from the specific source and target, while maintaining the gain value distribution. We minimize Mean Squared Error (MSE) and fit with SGD until convergence.

Before we present the results, it would be beneficial to give some intuition on MSE. As MSE is the square of the error, an average MSE of 4, means that the average prediction was accurate up to 2 ($\sqrt{4}$) points on average or more accurate than that (as outliers have more effect on MSE than on the average error rate).

We obtain an MSE of $4.2$ on the in-house models/targets experiment (3.3), versus $9.6, \sigma = 0.9$ when the gains are shuffled. Thus, knowing the source id and the target id provides significant information about the expected gain, with no need of keeping a designated parameter for each source-target combination. We further compare this regressor with two other regressors, one that considers only base model information ($\widehat{gain}(base_i, target_j) = b_i$) and one that considers only target related information ($\widehat{gain}(base_i, target_j) = t_j$). The MSE fit is $10.4$ and $8.2$, respectively, compared to $10.8, \sigma = 0.4$ on shuffled gains. This suggests both the source and the target impact the intertraining gain, with potentially somewhat stronger influence by the target.

In conclusion, our observations suggest that when considering intertraining potential, loosely speaking it is enough to consider two separate issues – (i) choosing a base model; and (ii) determining the expected effect on the target.

## 6 Factors Contributing to Gains

So far, we showed the effects of the intermediate models are largely separate from those of the target. We proceed to examine how specific factors in each contribute to intertraining gains.

### 6.1 Source and target sizes

Following the above observations, we ask what makes a base model good, or a target sensitive?

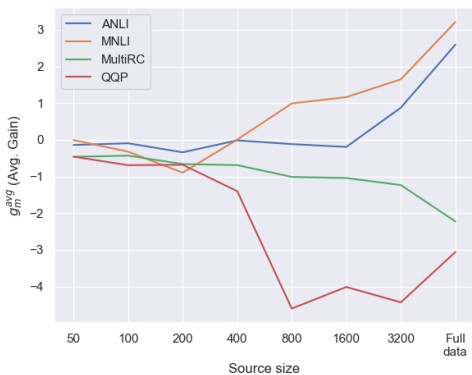

Figure 3: For 'good' sources the average gain increase as the source training size increases, while for 'bad' sources it decreases.

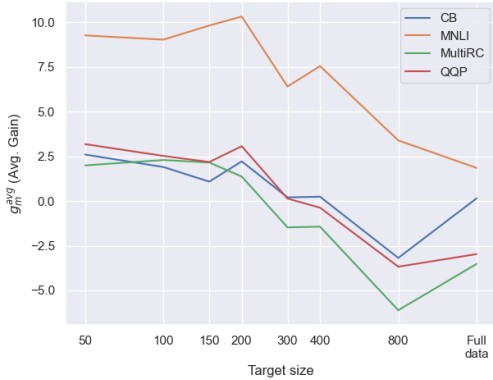

Figure 4: The average gain across targets decreases as the target training size increases.

We examine the effects of architecture (§6.3) and source score (§6.2), but start by examining the data sizes effect on the gain: First, we identify effects of the datasets sizes in controlled experiments. Next, we assess the extent to which the effect of dataset size is evident in previous experiments. For more related analysis we refer to Phang et al. (2018a); Pruksachatkun et al. (2020).

**Effect of dataset size.** We first control the source dataset train size. We create intermediate models on 4 source datasets – the top 2 and lowest 2 in-house models, according to the static ranking (§4.2). For each, we limit the training to $50 - 3200$ samples and use the General group datasets as targets. Evidently, for good sources (ANLI, MNLI), more training data yields greater gains (Fig. 3). However, additional data decreases performance for bad sources (MultiRC, QQP). We conclude that the amount of source data amplifies the effect determined by the type of data.

We experiment on the effect of target size, using General sources and General targets with train data of more than 1600 examples. We limit the target train sizes to between $50$ – namely, few-shot setting – and $1600$. As depicted in Fig. 4, the gain decreases as the target training size increases, implying larger potential for intertraining when the target training data is limited. Interestingly, for 3 targets we see positive gains on small datasets, which drops to negative gain as training data increases, and then seem to rise again towards zero. This 'U-shape' effect hints at two competing effects that should be better examined in future work (c.f. App. H).

**Training size effects in practice.** We examine whether the effect above is strong in comparison to other unknown factors. Considering the in-house models/targets experiment, the Pearson Correlation between source training size and source average gain is 0.75. Out of the 5 largest sources (ESNLI, MNLI, QQP, ANLI, and QNLI), 3 are also the source tasks with the highest average gain (MNLI, ANLI and ESNLI) and QQP is the source dataset with the second-lowest gain (negative). This is in line with our previous observation that additional source training data magnifies the positive or the negative intertraining effect of the source data.

We observe no significant correlation between target training size and target average gain, where the average is taken across sources. Still, targets with small training data seem to be more sensitive to intertraining. Specifically, the 5 targets with the smallest training data (CB, CoPA, WSC, WNLI, RTE) are also those for which we observe the maximal gain difference across all sources.

## 6.2 Similar Source Score, Different Gain

One can expect that two models with similar performance on a source dataset would also have similar intertraining gains. Our results suggest otherwise. We finetune 20 models over MNLI source and use them as intermediate models on the General target group. We compare the scores on the source test data to the average score obtained on the target datasets. Source task scores vary between 86.5 and 87.5 while General target average varies between 74.5 and 79, without correlation (c.f. App. I).

McCoy et al. (2019) found that finetuned models that show similar performance on their test data, still have different performance on out-of-domain challenge sets, suggesting the models learnt different generalizations schemes. Juneja et al. (2022a) suggested that those models converged to different basins in the loss space. They tagged models from one basin that tend to generalize better as *good*, and the rest as *bad*. We check whether good models are also better intermediate models for other tasks. We took their BERT models finetuned on MNLI as intermediate models – 12 good and 12 bad models – and used the General datasets as targets, finding that good models are indeed better for intertraining (3.65 avg. gain for good vs. 2.16 for bad).

The differences discussed above are due to using different seeds. In App. I.1 we show that hyperparameter selection can also impact the quality of an intermediate model, regardless of the score observed on the source test data.

In summary, similar training and/or similar performance on the source test, do not translate to similar intertraining gains on new target tasks.

## 6.3 Effect of different architectures

We validate our main conclusions across different architectures. To that end, we repeat the in-house models/targets experiment with BERT (Devlin et al., 2019) and T5 (Raffel et al., 2020) architectures. (see full tables in App. J).

We start by showing that the loose source-target

coupling holds across architectures. We then show that different source datasets rank differently across architecture, but that target sensitivity is similar.

To show the source-target independence, we repeat the regression fit (§5). As before, the fit explains each architecture's gains much better than when data is shuffled (BERT MSE $10.5$, random $30.1, \sigma = 4.17$; T5 MSE $8.11$, random $13.51, \sigma = 1.5$). Neither the average gain of intermediate models - trained over various sources, nor the average gain for target tasks, correlate across different architectures. However, the target sensitivity, measured by max-gain, is correlated across all architectures (pairwise Pearson $0.6 - 0.94, p < 0.05$). Thus, although the source-target decoupling and the target sensitivity are shared across architectures, a source dataset that produces high gains in one architecture might not do so in another.

A notable exception is the MNLI source dataset which achieves the highest gain in all three architectures. Possibly, some data sources provide a strong intermediate model regardless of PT, with MNLI as a prominent example.

## 7  Related Work

Various works use intertraining, often following the assumption of task alignment necessity (Ein-Dor et al., 2022; Don-Yehiya et al., 2022a; Awasthy et al., 2020), namely, that the source acts as weak labeled data (Shnarch et al., 2018; Yu et al., 2021). While we consider intertraining through finetuning, adaptation to the target (Shnarch et al., 2022) or the domain (Gururangan et al., 2020) was also suggested. Such adaptation may be applied to any base model, and is complementary to the choice among base models. The need for alignment was also previously debated in the context of pretraining tasks (Krishna et al., 2021; Rothe et al., 2021; Zhang et al., 2020; Ram et al., 2021).

The properties of intertraining were studied in other contexts. Phang et al. (2018a) suggested the intertraining scheme. Pruksachatkun et al. (2020) probed the linguistic knowledge changes after intertraining, noting correlations to some target tasks, and hypothesized why some source tasks are good for intertraining. Mosbach et al. (2020); Chang and Lu (2021) replicated the existence of good sources, but rejected the hypothesis. Others tried to find which tasks have an affinity to each other (Poth

et al., 2021; Bassignana et al., 2022a,b; Vu et al., 2020), or when to prefer multitask (Weller et al., 2022). We study a much larger number of sources and targets, aiming to describe their natural distribution (c.f. §9) and also find that while specific choice may help, given enough models, it is safe to identify the models that just excel generally.

Recent work focuses on fusing multiple base models rather than picking just one (Choshen et al., 2022; Matena and Raffel, 2021; Wortsman et al., 2022; Yadav et al., 2023). We expect our understanding to aid in choosing a subset of models to fuse as well.

Multi-task learning is another related field. It studies finetuning on different tasks at once (Aribandi et al., 2021; Aghajanyan et al., 2021) and recently also a way to recycle models to do so was proposed (Don-Yehiya et al., 2022b). In contrast to our analysis, similarity between tasks aids multi-task learning (Abnar et al., 2021).

Deciding which information should accompany publications is an active research field, covering datasets (Holland et al., 2018; Gebru et al., 2021), human annotation sheets (Shimorina and Belz, 2022), and models (McMillan-Major et al., 2021; Mitchell et al., 2019). Our work proposes to report upon sharing a model the aspects shown to affect its quality, such as $LP\,(m, MNLI)$. For datasets, we propose to report intertraining sensitivity.

## 8  Practical recommendations

Based on the observations (§4) and analysis (§5), we propose a methodology for efficiently utilizing intertraining in real-world settings. We suggest to collaboratively rank all available base models for intertraining and to utilize this list whenever intertraining is applied to a new target.

**New model.** When a new model becomes available, we encourage practitioners to assess and share its quality. This can be done efficiently by linear probing on MNLI (§4.2) or comprehensively (App. §F) by finetuning on various datasets.

We created statically ranked lists for RoBERTa-base and T5-small in App. §E. Moreover, we apply our methods to many architectures and 36 test sets in an updating site , reporting the best models.

**New target.** When considering intertraining on a new task, we recommend checking the target dataset sensitivity, and then choosing the base model. Since the model's rank hardly depends on the target dataset, we suggest using the static ranking. Specifically, we propose to finetune the top-ranked model, and compare the results to those obtained when finetuning the respective PT model. Assuming the gain justifies it, one should consider a few additional intermediate models, in descending order, according to the allocated resources.

## 9 Discussion

In §8, we highlighted our practical recommendations for intertraining. Those, together with a systematic analysis of what affects intertraining, cover our main contributions. We hope this analysis would also aid future work on interactions between datasets; intertraining practices; and reusing fine-tuned models.

Our experiments mainly characterize what is probable rather than what is possible. We do not create specific models or aim to improve a specific task. Instead, we investigate what is likely to be found in typical practical settings. Accordingly, our findings are probabilistic in nature: Most models are not beneficial as intermediate models, but there are enough that are. Mostly, beneficial models are beneficial for many targets.

As a side effect, we do identify specific strong source models. MNLI was already considered a beneficial source dataset (Phang et al., 2018a), a finding which we corroborate in a comprehensive manner. Furthermore, when considering off-the-shelf models we find models that outperform it (e.g., STS-B based for RoBERTa, and Quora for T5). To facilitate finding additional and better base models we will continuously report in the website website the best models per architecture.

## 10 Limitations

One obvious limitation is that our work is empirical in nature. As such, we report observations, sampled or common, with no theoretical guarantees, and one should recognize the existence of exceptions. Specifically, even though we have not observed it – there might exist target tasks that benefit greatly from a certain type of intermediate models; or intermediate models that help greatly in many targets while degrading performance in others.

Moreover, while testing 22 source datasets, many of which previously untested, we did not find a new top source for intertraining. The best one we found for RoBERTa was already known to be good (MNLI; Phang et al., 2018b; Pruksachatkun et al., 2020). With that, by checking dozens of off-the-shelf models, we did uncover new intermediate models that seem to outperform MNLI (e.g., STS-B for RoBERTa and QQP for T5 – c.f. App. §E). More work is needed to assess the potential intertraining gain of the available datasets and models.

We ran thousands of finetuning experiments, spanning a vast number of tasks and base models. Thus, while we believe it is unlikely that reproducing our experiments will result in different outcomes, the large scale of our experiments places a heavy burden on trying to replicate our results. Moreover, the off-the-shelf models used in our experiments might not be hosted publicly in the future.

Another limitation is that we could not upload all of the models to a shared location. This project was very computationally demanding, but more than that, it was demanding in terms of disk space, hence we had to delete many models along the way.

Finally, for practical reasons, our results are limited to classification tasks in English. We hope that future work will aim to test our conclusions beyond this scope. Overall, in the space of classification, we see our results as robust, testing on 22 datasets (about double the amount of previous works (Pruksachatkun et al., 2020)). We hope the diversity of targets brings large enough differences between datasets that the results would hold in other scopes.

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

## A Task, Domain and Dataset

A *task* is defined by the input and the output. The input in our context is a text instance. The output could be, e.g., positive/negative/neutral for a sentiment analysis task, entailed/not-entailed for a textual entailment task, etc.

A *domain* is the type of text found in the examples, regardless of the labels. For example, a domain may be financial or comments for twitter.

A *dataset* for our purpose is a set of examples and their labels, divided into train, dev, and test folds. Being such, each dataset has a domain (characterizing its examples) and a task (for which its labels are annotated). Hence, we consider a subset of a dataset as an another dataset. Note that in the literature those terms are often not well defined and may even be interchangeable (Wang et al., 2019b).

## B Hyperparameters

For RoBERTa, we train for 10 epochs with linear learning rate 5e-5 with warm-up of 0.6% of training, batch size of 256, early stop epsilon 0.001 accuracy points, patience of $20 \times 50 \times 256$ examples, validate every $50 \times 256$ examples, optimizer ADAMW(Loshchilov and Hutter, 2019), with weight decay 0.01 or 0. For BERT-base-uncased we use 2e-5 learning rate and never use decay. For T5 we use 1e-4 learning rate and train until early stopping occurs. We used A100 and V100 GPUs. Finetuning times vary, but all end within a couple of hours, most in less than an hour, some up to 8 hours.

## C Datasets used

All datasets could be downloaded from hugging-face datasets. As we used groups of datasets we report here the full list of datasets they contain.

**General** GLUE: CoLA (Warstadt et al., 2019), SST2 (Socher et al., 2013), MRPC (Dolan and Brockett, 2005), QQP (data.quora.com/First-Quora-Dataset-Release-Question-Pairs), MNLI (Williams et al., 2018), QNLI Rajpurkar et al. 2016, RTE (Dagan et al., 2005; Bar-Haim et al., 2006; Giampiccolo et al., 2007; Bentivogli et al., 2009), WNLI (Levesque et al., 2011)

SuperGLUE: BoolQ (Clark et al., 2019), CB (de Marneffe et al., 2019), CoPA (Roemmele et al.,

2011), MULTIRC (Khashabi et al., 2018), WIC (Pilehvar and Camacho-Collados, 2019), WSC (Levesque et al., 2012)

**NLI:** MNLI (Williams et al., 2018), QNLI Rajpurkar et al. 2016, RTE (Dagan et al., 2005; Bar-Haim et al., 2006; Giampiccolo et al., 2007; Bentivogli et al., 2009), WNLI (Levesque et al., 2011), ESNLI (Camburu et al., 2018), adversarial NLI (Nie et al., 2020).

**Twitter:** EmoInt (Mohammad and Bravo-Marquez, 2017), Emoji (Barbieri et al., 2018), Irony (Van Hee et al., 2018), OffenseEval (Zampieri et al., 2019), HatEval (Basile et al., 2019) , Sentiment Analysis (Rosenthal et al., 2017)

Whenever the test set is held out (such as is GLUE and SuperGLUE), we extracted 1K or 10% of the training examples as test set, the smaller. We did not experiment with the small Stance (Mohammad et al., 2016) Twitter dataset originally found in TweetEval(Barbieri et al., 2020) to reduce noise. In the T5 experiment (§E) we used stance datasets as well, to have a large group. For MNLI we use the mismatched validation set as a test and the matched as a validation set.

## D In-house models/targets additional information

We report in Table 3 the score of finetuning RoBERTa without intertraining.

We also report the standard deviation for each cell in the experiment, i.e., taking into account differences due to finetuning the intermediate model and target dataset in figure 5. For each seed, we finetune the PT over the source dataset to produce the base model, and use it to finetune the target task. It means that each seed utilises a different base model. Note that §6.2 suggest different seeds may create base models with different quality. Note that to assess the variability of the averages reported in the main text (§4) the Standard Error of the Mean is required, this is the STD divided by the square of the number of seeds $SE = STD/\sqrt{5}$.

## E Models in the wild

### E.1 Models used

We collected manually 66 RoBERTa-base models. From their names most were finetuned from vanila

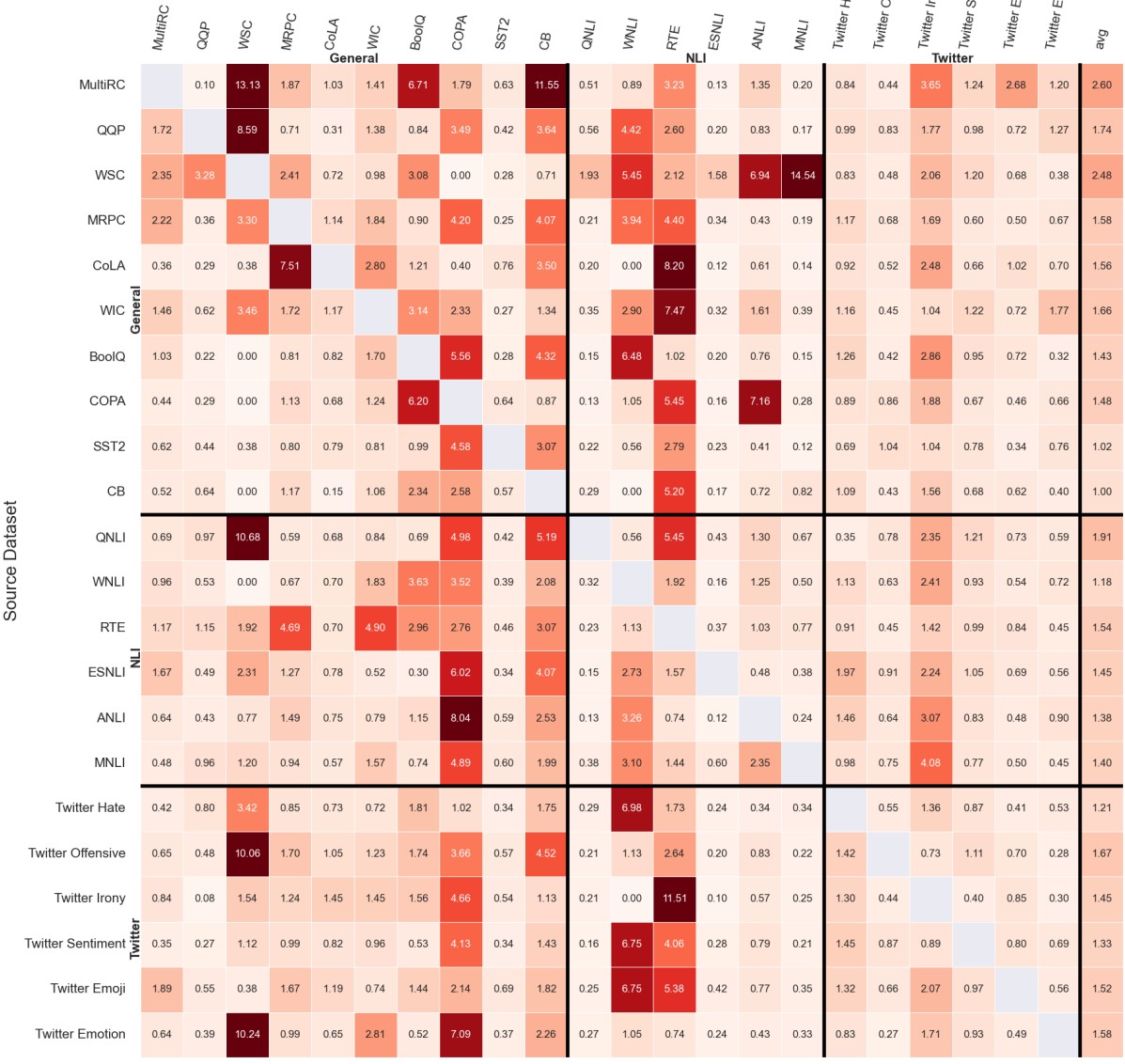

Figure 5: Standard deviation of in-house models/targets experiment. Rows correspond to intermediate models, generated based on 22 source datasets from the General, NLI and Twitter groups. Columns correspond to the same datasets, now being used as target datasets. Each value indicates standard deviation over 5 seeds.

| Dataset | Mean | Std |
|---|---|---|
| MultiRC | 61.07 | 2.01 |
| QQP | 90.92 | 0.29 |
| WSC | 63.46 | 0.00 |
| MRPC | 87.70 | 0.95 |
| CoLA | 83.11 | 1.34 |
| WIC | 65.55 | 2.32 |
| BoolQ | 77.09 | 3.19 |
| COPA | 49.00 | 4.90 |
| SST2 | 93.81 | 0.26 |
| CB | 70.36 | 3.11 |
| QNLI | 92.28 | 0.48 |
| WNLI | 56.34 | 0.00 |
| RTE | 72.42 | 0.93 |
| ESNLI | 91.05 | 0.18 |
| ANLI | 51.67 | 0.36 |
| MNLI | 87.07 | 0.23 |
| Twitter Hate | 52.30 | 1.03 |
| Twitter Offensive | 84.67 | 0.41 |
| Twitter Irony | 70.84 | 2.53 |
| Twitter Sentiment | 70.59 | 0.34 |
| Twitter Emoji | 46.32 | 0.56 |
| Twitter Emotion | 82.08 | 0.58 |

Table 3: Scores of finetuning RoBERTa without inter-training.

RoBERTa, but a few were trained from scratch. They vary in hyperparameters and training details, just as one can expect training approaches to vary between different researchers. We supply the full list of models in Tables 4,5 for RoBERTa and T5 respectively.

### E.2 Results and discussion

We report the full results in Fig. 6. T5 models are reported in §7 (off-the-sheld are on Twitter datasets only, as most of General and NLI were part of T5's pretraining, in-house use t5.1.1). Seemingly, some traits of the training that we did not account for are important. It is exemplified by models that are associated with the same datasets but differ in their gains. For example cross-encoder implementations outperform other models and sentence-transformers underperform them.

Notably, most models are not useful for inter-training. Still, many models do.

The best model on average is MUPPET (Agha-janyan et al., 2021) which is a massive multitask learning approach. However, our results show

that finetuning only on STS-B (rather than on 40 datasets) yielded similar results. We note that unlike MNLI which is known to be a good source (Phang et al., 2018a), STS-B was previously only considered as a target task only. The results suggest, it might be a good source.

We gathered the models by manually searching for 'RoBERTA-base' models, ignoring ones that were working on languages other than English. It is possible we have missed models that did not clearly state their architecture as part of the model name. We are already aware of such models, for which the PT is not deducible from their title, for example those lately released by Juneja et al. (2022b).

### F Rank by Average over Targets

In §4.2 we show training one Linear Probe is enough to rank base models. Although tested on a large number of target datasets, presumably, this method does not always achieve accurate predictions. For example, the target domain might be so different that MNLI would not be relevant. As a more accurate alternative, one can use several datasets to provide a more reliable picture. We show that an average of finetuning gains over different datasets is a reliable way for choosing a base model. As in the simpler case of LP, this supports the decoupling.

We report in Table 6 the lost gain when choosing the highest models, ranked by average gain over the General group. This ranking method generalize well; The 1 or 3 best-ranked models are close to the best possible model overall for each target. For example, only 2 targets lose more than 1 point by choosing the top 3 models.

Practically, we suggest to rank either in this method or by LP. If some use one method and others choose another it might be hard to compare the two rankings. Thus, we report that in our experiments the best predictor of the average score by the LP score is $g_m^{avg} = LP(m, MNLI) \cdot 0.0822 - 0.940$

### G Symmetry metric

To measure symmetry of a matrix M, we decompose it to it its symmetrical and skew symmetrical parts: $M = S + V$ where $S = \frac{1}{2}(M + M^T)$ and $V = \frac{1}{2}(M - M^T)$. S is symmetrical: $S = S^T$ and V is skew-symmetrical: $V = -V^T$. The measure

| | set | name | avg. gain over General | LP gain over MNLI |
|---|---|---|---|---|
| 0 | imdb _1 | aychang/roberta-base-imdb | -5.91 | -12.62 |
| 1 | sentence_4 | sentence-transformers/stsb-roberta-base | -5.84 | 2.59 |
| 2 | models_1 | textattack/roberta-base-ag-news | -5.73 | -17.09 |
| 3 | twitter_10 | lucaordronneau/twitter-roberta-base-sentiment-... | -5.71 | -9.76 |
| 4 | sentence_5 | sentence-transformers/roberta-base-nli-stsb-me... | -5.58 | 2.59 |
| 5 | finance_0 | zhayunduo/roberta-base-stocktwits-finetuned | -4.49 | -12.60 |
| 6 | sentence_2 | sentence-transformers/msmarco-roberta-base-v3 | -4.17 | -8.82 |
| 7 | twitter_8 | cardiffnlp/twitter-roberta-base-emotion | -4.17 | 1.00 |
| 8 | sentence_1 | sentence-transformers/roberta-base-nli-mean-to... | -4.07 | 5.47 |
| 9 | qa_3 | navteca/roberta-base-squad2 | -4.04 | 4.03 |
| 10 | quora _0 | cross-encoder/quora-roberta-base | -3.61 | 8.35 |
| 11 | scratch_0 | neoyipeng/twitter-roberta-base-sentiment-mlm-c... | -3.51 | -3.13 |
| 12 | models_5 | neoyipeng/twitter-roberta-base-sentiment-mlm-c... | -3.51 | -3.13 |
| 13 | sentence_0 | sentence-transformers/nli-roberta-base | -3.50 | 5.47 |
| 14 | models_8 | cointegrated/roberta-base-formality | -3.29 | -4.96 |
| 15 | legal_1 | saibo/legal-roberta-base | -3.20 | -1.76 |
| 16 | twitter_12 | cardiffnlp/twitter-roberta-base-stance-abortion | -2.95 | -8.89 |
| 17 | sst2 _0 | Bhumika/roberta-base-finetuned-sst2 | -2.77 | -3.00 |
| 18 | models_14 | cestwc/roberta-base-unigram-ternary | -2.73 | -8.39 |
| 19 | models_11 | mariagrandury/roberta-base-finetuned-sms-spam-... | -2.67 | -5.90 |
| 20 | qa_1 | nlpconnect/roberta-base-squad2-nq | -2.57 | 3.67 |
| 21 | twitter_13 | bdotloh/twitter-roberta-base-finetuned-twitter... | -2.47 | -3.12 |
| 22 | models_0 | textattack/roberta-base-CoLA | -2.38 | -2.47 |
| 23 | twitter_6 | cardiffnlp/twitter-roberta-base-dec2021 | -2.31 | -11.45 |
| 24 | twitter_5 | cardiffnlp/twitter-roberta-base-mar2022 | -2.21 | -5.38 |
| 25 | quora _1 | navteca/quora-roberta-base | -2.13 | 8.35 |
| 26 | models_16 | hoanhkhoa/roberta-base-finetuned-ner | -2.12 | -6.82 |
| 27 | twitter_3 | cardiffnlp/twitter-roberta-base-2021-124m | -2.05 | -5.30 |
| 28 | twitter_11 | cardiffnlp/twitter-roberta-base-stance-climate | -1.98 | -6.29 |
| 29 | legal_0 | akdeniz27/roberta-base-cuad | -1.90 | -8.57 |
| 30 | models_13 | cardiffnlp/twitter-roberta-base-stance-feminist | -1.86 | -4.25 |
| 31 | imdb _2 | aypan17/roberta-base-imdb | -1.86 | -3.50 |
| 32 | models_2 | ghanashyamvtatti/roberta-fake-news | -1.81 | -12.47 |
| 33 | scratch_1 | neoyipeng/twitter-roberta-base-sentiment-mlm-skep | -1.71 | -8.73 |
| 34 | models_12 | surrey-nlp/roberta-base-finetuned-abbr | -1.65 | -0.66 |
| 35 | twitter_4 | cardiffnlp/twitter-roberta-base-emoji | -1.63 | -2.76 |
| 36 | models_4 | textattack/roberta-base-rotten-tomatoes | -1.61 | 3.73 |
| 37 | legal_2 | marshmellow77/roberta-base-cuad | -1.53 | -8.57 |
| 38 | legal_3 | Rakib/roberta-base-on-cuad | -1.33 | -2.48 |
| 39 | twitter_7 | cardiffnlp/twitter-roberta-base-sentiment | -1.24 | 3.03 |
| 40 | twitter_9 | cardiffnlp/twitter-roberta-base | -1.16 | -1.49 |
| 41 | models_15 | thatdramebaazguy/roberta-base-wikimovies | -1.11 | -1.76 |
| 42 | finance_1 | vanadhi/roberta-base-fiqa-flm-sq-flit | -1.09 | -1.00 |
| 43 | models_3 | allenai/reviews | -1.01 | -1.17 |
| 44 | models_7 | princeton-nlp/sup-simcse-roberta-base | -0.99 | 4.26 |
| 45 | mrpc _1 | ji-xin/roberta | -0.94 | -1.67 |
| 46 | twitter_1 | cardiffnlp/twitter-roberta-base-offensive | -0.91 | -2.79 |
| 47 | sst2 _1 | textattack/roberta-base-SST-2 | -0.84 | 3.83 |
| 48 | sentence_3 | sentence-transformers/stsb-roberta-base-v2 | -0.80 | 1.68 |
| 49 | twitter_2 | cardiffnlp/twitter-roberta-base-irony | -0.69 | 0.46 |
| 50 | imdb _0 | textattack/roberta-base-imdb | -0.40 | -3.64 |
| 51 | models_6 | VictorSanh/roberta-base-finetuned-yelp-polarity | -0.25 | -0.66 |
| 52 | models_10 | gargam/roberta-base-crest | -0.12 | -4.21 |
| 53 | twitter_0 | bhadresh-savani/roberta-base-emotion | -0.05 | -4.61 |
| 54 | qa_4 | shahrukhx01/roberta-base-boolq | 0.25 | 10.42 |
| 55 | mrpc _0 | textattack/roberta-base-MRPC | 0.39 | 7.27 |
| 56 | nli_3 | textattack/roberta-base-RTE | 0.42 | 14.82 |
| 57 | nli_2 | mujeensung/roberta-base_mnli_bc | 0.48 | 23.43 |
| 58 | qa_0 | deepset/roberta-base-squad2-covid | 0.50 | -1.11 |
| 59 | qa_2 | csarron/roberta-base-squad-v1 | 0.99 | 3.38 |
| 60 | stsb _0 | textattack/roberta-base-STS-B | 1.05 | 14.66 |
| 61 | models_9 | textattack/roberta-base-QNLI | 1.10 | 6.49 |
| 62 | nli_0 | textattack/roberta-base-MNLI | 2.09 | 34.39 |
| 63 | nli_1 | cross-encoder/nli-roberta-base | 2.77 | 34.09 |
| 64 | stsb _1 | cross-encoder/stsb-roberta-base | 2.82 | 30.19 |
| 65 | multitask_0 | facebook/muppet-roberta-base | 3.00 | 31.58 |

Table 4: RoBERTa models we used, collected from Hugging Face models hub. Models sorted by average gain over the General targets.

Target Dataset

| Source Dataset | BoolQ | CB | CoLA | COPA | MNLI | MRPC | MultiRC | QNLI | QQP | RTE | SST2 | WiC | WNLI | WSC | avg |
|---|---|---|---|---|---|---|---|---|---|---|---|---|---|---|---|
| Models 11 | -15.11 | -28.57 | -14.29 | 4.00 | -51.45 | -17.16 | -3.16 | -30.77 | -14.95 | -27.08 | -14.56 | -20.85 | 0.00 | 0.00 | -16.71 |
| Imdb 1 | -7.34 | -14.29 | -5.56 | -9.00 | 0.26 | -1.96 | -3.16 | -0.73 | -0.17 | -17.69 | -1.26 | -7.68 | -8.45 | -5.77 | -5.91 |
| Sentence 4 | -4.28 | -1.79 | -2.97 | -3.00 | -1.87 | -5.64 | 2.15 | -0.33 | -3.45 | -20.22 | -0.34 | -6.27 | -33.80 | 0.00 | -5.84 |
| Models 1 | -12.84 | 0.00 | -8.05 | 2.00 | -1.60 | -13.48 | -2.50 | -1.94 | -1.13 | -23.10 | -2.41 | -10.97 | -4.23 | 0.00 | -5.73 |
| Twitter 10 | -5.47 | 1.79 | -2.40 | -9.00 | -1.20 | -6.13 | -1.24 | -1.30 | -2.03 | -20.94 | -0.92 | -5.17 | 0.00 | -25.96 | -5.71 |
| Sentence 5 | -3.67 | -1.79 | -2.01 | -2.00 | -2.72 | -4.66 | 2.15 | -0.02 | -3.45 | -20.22 | -0.34 | -5.64 | -33.80 | 0.00 | -5.58 |
| Finance 0 | -8.29 | -3.57 | -4.22 | 2.00 | -2.66 | -14.22 | -2.00 | -1.46 | -0.48 | -21.30 | -0.11 | -6.43 | 2.82 | -2.88 | -4.49 |
| Sentence 2 | -2.57 | -1.79 | -1.92 | -4.00 | -2.23 | -2.70 | 0.74 | -0.48 | 0.18 | -14.44 | 1.15 | -5.02 | -25.35 | 0.00 | -4.17 |
| Twitter 8 | -5.41 | -1.79 | -2.01 | -10.00 | -0.34 | 0.49 | -3.80 | -0.51 | 0.13 | -20.58 | -0.80 | -1.10 | -12.68 | 0.00 | -4.17 |
| Sentence 1 | -2.57 | 5.36 | -2.97 | 4.00 | -3.10 | -3.19 | -3.16 | -0.16 | -1.18 | -15.52 | -0.11 | -3.45 | -30.99 | 0.00 | -4.07 |
| QA 3 | -15.11 | 0.00 | -6.14 | 4.00 | -7.11 | -1.96 | -3.16 | 0.07 | -1.27 | -25.63 | -0.57 | 0.31 | 0.00 | 0.00 | -4.04 |
| Quora 0 | -11.44 | -21.43 | -4.12 | -8.00 | 0.66 | 1.47 | 2.81 | -0.20 | 1.56 | 1.08 | -1.61 | 1.10 | -2.82 | -9.62 | -3.61 |
| Models 5 | -5.87 | 3.57 | -1.25 | 5.00 | -10.75 | 0.74 | -3.16 | -5.42 | -5.45 | -17.33 | -0.23 | -3.76 | -4.23 | -0.96 | -3.51 |
| Scratch 0 | -5.87 | 3.57 | -1.25 | 5.00 | -10.75 | 0.74 | -3.16 | -5.42 | -5.45 | -17.33 | -0.23 | -3.76 | -4.23 | -0.96 | -3.51 |
| Sentence 0 | -1.80 | 7.14 | -2.30 | 4.00 | -3.82 | -1.47 | -3.16 | 0.09 | -3.37 | -19.86 | 0.11 | -3.45 | -21.13 | 0.00 | -3.50 |
| Models 8 | -0.86 | -1.79 | -0.86 | 4.00 | 0.27 | 0.00 | 2.06 | -0.51 | -0.59 | -18.77 | -0.23 | -2.82 | 0.00 | -25.96 | -3.29 |
| Legal 1 | -4.74 | 0.00 | -1.05 | -8.00 | -10.55 | 1.23 | -0.21 | -4.56 | -5.76 | -7.58 | -1.26 | -2.35 | 0.00 | 0.00 | -3.20 |
| Twitter 12 | -6.33 | -1.79 | -1.25 | 0.00 | -1.75 | 0.74 | -3.16 | -0.73 | -0.23 | -15.52 | -0.23 | -2.51 | -7.04 | 0.00 | -2.95 |
| SST2 0 | -2.66 | 0.00 | -1.73 | 2.00 | -0.24 | -0.49 | 2.02 | -0.40 | -0.15 | -15.88 | -0.46 | -5.33 | -15.49 | 0.00 | -2.77 |
| Models 15 | -5.69 | -3.57 | -8.15 | 4.00 | -3.51 | -0.25 | 0.68 | -1.46 | -2.43 | -13.36 | -0.92 | -3.61 | 0.00 | 0.00 | -2.73 |
| Models 12 | -0.55 | -1.79 | -4.51 | 4.00 | -1.12 | -0.49 | 2.33 | 0.33 | -3.21 | -3.25 | 0.57 | -2.82 | 0.00 | -26.92 | -2.67 |
| QA 1 | 1.90 | 0.00 | -4.89 | 4.00 | 0.36 | -0.98 | -0.72 | 0.18 | -9.96 | -27.08 | -0.46 | 1.72 | 0.00 | 0.00 | -2.57 |
| Twitter 13 | -4.98 | 0.00 | -2.01 | -2.00 | -2.23 | -0.49 | -3.16 | -1.24 | -1.27 | -9.39 | -0.46 | -1.72 | -4.23 | 0.00 | -2.47 |
| Models 0 | 2.66 | -1.79 | 0.96 | 4.00 | -1.37 | 3.19 | 3.05 | -0.15 | -2.13 | -15.16 | -0.23 | -3.76 | -19.72 | -2.88 | -2.38 |
| Twitter 6 | -0.15 | -3.57 | -1.15 | 5.00 | -1.00 | 0.00 | 0.02 | -1.04 | -0.17 | -10.83 | 0.11 | -1.41 | -18.31 | 0.00 | -2.31 |
| Twitter 5 | -2.05 | -3.57 | -2.21 | 5.00 | -0.54 | 1.72 | -0.66 | -1.43 | -1.27 | -7.94 | 0.57 | -0.31 | -18.31 | 0.00 | -2.21 |
| Quora 1 | -10.28 | -8.93 | -4.12 | -5.00 | 0.67 | 0.49 | 1.79 | -0.20 | 1.09 | 2.89 | -1.26 | 0.31 | 1.41 | -8.65 | -2.13 |
| Models 17 | -9.57 | -1.79 | -1.15 | -4.00 | 0.03 | 1.96 | -3.16 | 0.13 | -0.83 | -6.50 | -0.46 | -2.98 | -1.41 | 0.00 | -2.12 |
| Twitter 3 | -0.03 | -3.57 | -1.15 | 3.00 | -0.27 | 1.23 | -0.87 | -1.21 | -0.03 | -9.39 | -0.11 | -0.78 | -15.49 | 0.00 | -2.05 |
| Twitter 11 | -5.66 | -1.79 | -2.11 | 4.00 | -1.57 | 1.23 | -1.24 | -0.82 | -1.63 | -16.25 | 0.11 | -2.04 | 0.00 | 0.00 | -1.98 |
| Legal 0 | -5.20 | 1.79 | -3.45 | 4.00 | -0.02 | -1.72 | -3.16 | -1.23 | 0.04 | -7.94 | -0.92 | -8.78 | 0.00 | 0.00 | -1.90 |
| Models 14 | -7.03 | -1.79 | -1.15 | 4.00 | -1.67 | 0.98 | 1.82 | -1.03 | -1.67 | -16.97 | -0.80 | -0.78 | 0.00 | 0.00 | -1.86 |
| Imdb 2 | -0.83 | 3.57 | -3.26 | 4.00 | 0.21 | 1.47 | 2.50 | 0.04 | -0.20 | -5.42 | 0.11 | -1.25 | 0.00 | -26.92 | -1.86 |
| Models 2 | -2.84 | -3.57 | -1.15 | 4.00 | 0.36 | 0.74 | 2.02 | 0.00 | -0.30 | -13.72 | -0.80 | -2.98 | -7.04 | 0.00 | -1.81 |
| Scratch 1 | -6.64 | 0.00 | -3.16 | 4.00 | -0.49 | 0.00 | 0.14 | -1.45 | -0.35 | -14.44 | -0.11 | -1.41 | 0.00 | 0.00 | -1.71 |
| Models 13 | -0.43 | -1.79 | -0.38 | -6.00 | 0.23 | 2.45 | 2.29 | -1.04 | -1.66 | -5.78 | 0.00 | -1.41 | -2.82 | -6.73 | -1.65 |
| Twitter 4 | -3.58 | 0.00 | -3.07 | 4.00 | -0.28 | 0.98 | -3.16 | -0.40 | 0.18 | -10.11 | 0.23 | -0.63 | -7.04 | 0.00 | -1.63 |
| Models 4 | 1.10 | 0.00 | -1.15 | -15.00 | -1.80 | 2.94 | -0.29 | -0.79 | -1.32 | -4.69 | 1.15 | -2.66 | 0.00 | 0.00 | -1.61 |
| Legal 2 | -5.20 | 1.79 | -3.64 | 4.00 | 0.22 | 0.00 | -3.16 | -1.30 | 0.04 | -7.94 | -0.92 | -5.33 | 0.00 | 0.00 | -1.53 |
| Legal 3 | -0.37 | 1.79 | -3.74 | -5.00 | 0.62 | 1.47 | 1.88 | -0.46 | -0.95 | -14.44 | 0.11 | 0.47 | 0.00 | 0.00 | -1.33 |
| Twitter 7 | -5.54 | 0.00 | -2.01 | 8.00 | -0.13 | 1.72 | -2.54 | -0.79 | -1.47 | -10.47 | -1.26 | -2.82 | 0.00 | 0.00 | -1.24 |
| Twitter 9 | -3.73 | 0.00 | -0.77 | -4.00 | -0.72 | 0.25 | 0.04 | -0.71 | -0.95 | -5.05 | -0.23 | -0.31 | 0.00 | 0.00 | -1.16 |
| Models 16 | 1.31 | 1.79 | 0.10 | -12.00 | -0.85 | 0.49 | 2.64 | -0.44 | -2.72 | -3.97 | -0.80 | -1.10 | 0.00 | 0.00 | -1.11 |
| Finance 1 | 1.44 | -1.79 | -2.78 | 4.00 | -1.41 | 1.23 | 3.05 | 0.70 | 0.16 | 1.08 | -0.46 | 2.66 | 0.00 | -23.08 | -1.09 |
| Models 3 | -4.37 | -1.79 | -2.40 | 2.00 | -1.31 | -0.49 | 0.12 | -1.65 | -0.84 | -4.69 | 0.69 | 0.63 | 0.00 | 0.00 | -1.01 |
| Models 7 | 1.87 | 1.79 | -1.73 | -6.00 | -3.34 | 2.70 | 1.38 | -1.63 | -3.86 | -3.97 | 0.00 | -1.10 | 0.00 | 0.00 | -0.99 |
| MRPC 1 | 0.49 | 0.00 | -0.48 | 1.00 | 0.55 | 0.49 | 1.67 | 0.04 | -2.29 | -14.08 | -0.23 | -0.31 | 0.00 | 0.00 | -0.94 |
| Twitter 1 | -6.24 | -1.79 | -0.96 | 6.00 | -0.23 | 1.23 | 0.27 | -0.86 | -0.10 | -7.22 | -0.57 | -2.19 | 0.00 | 0.00 | -0.91 |
| SST2 1 | 1.50 | 1.79 | -0.67 | -5.00 | -0.15 | 2.21 | 3.09 | 0.05 | -1.96 | -9.75 | -0.46 | -2.35 | 0.00 | 0.00 | -0.84 |
| Sentence 3 | 1.77 | 5.36 | -0.38 | -1.00 | -0.23 | 1.72 | 3.44 | 0.53 | -2.73 | -13.72 | -0.23 | -1.57 | -4.23 | 0.00 | -0.80 |
| Twitter 2 | -3.61 | -3.57 | -0.14 | 4.00 | -0.29 | 1.47 | -0.71 | -0.14 | 1.10 | -6.50 | -0.34 | 1.10 | 0.00 | 0.00 | -0.69 |
| Imdb 0 | 1.07 | 3.57 | -0.86 | 1.00 | -0.95 | 2.94 | -3.16 | -0.27 | -0.17 | -6.14 | 0.00 | -2.66 | 0.00 | 0.00 | -0.40 |
| Models 6 | 2.69 | -1.79 | -0.48 | 3.00 | 0.54 | 3.19 | 1.88 | -0.07 | -0.29 | -4.33 | 0.11 | 0.47 | -8.45 | 0.00 | -0.25 |
| Models 10 | -2.20 | 1.79 | -2.21 | 4.00 | 0.44 | 1.72 | -3.16 | -0.79 | -0.01 | 2.53 | -0.34 | -3.45 | 0.00 | 0.00 | -0.12 |
| Twitter 0 | 0.95 | 1.79 | -0.29 | 4.00 | 0.48 | 0.25 | 2.39 | -0.07 | -0.22 | -6.86 | 0.34 | -2.51 | 0.00 | -0.96 | -0.05 |
| QA 4 | 3.82 | -3.57 | -1.53 | 5.00 | -1.71 | 0.98 | 3.82 | -0.37 | -0.37 | 1.08 | 0.00 | -1.25 | 0.00 | 0.00 | 0.25 |
| MRPC 0 | 2.72 | 0.00 | 0.19 | 4.00 | -1.83 | 3.92 | 3.40 | -0.64 | -1.89 | -5.42 | 0.46 | 0.47 | 0.00 | 0.00 | 0.39 |
| NLI 3 | 1.53 | 0.00 | -0.48 | 3.00 | -3.39 | 1.96 | 2.08 | 0.26 | -0.46 | 3.61 | -0.46 | -1.72 | 0.00 | 0.00 | 0.42 |
| NLI 2 | 1.07 | 5.36 | -3.16 | 2.00 | -0.20 | 0.25 | 1.16 | -0.11 | -0.31 | 5.05 | 0.46 | -1.10 | -2.82 | -0.96 | 0.48 |
| QA 0 | 1.93 | -1.79 | -0.38 | 4.00 | 0.15 | 1.72 | 0.35 | 0.16 | 0.03 | 0.36 | -0.92 | 1.41 | 0.00 | 0.00 | 0.50 |
| QA 2 | 0.52 | 3.57 | -1.53 | 5.00 | -0.02 | 2.45 | 1.01 | 0.60 | -0.47 | 1.44 | -0.34 | 1.57 | 0.00 | 0.00 | 0.99 |
| STS-B 0 | 1.74 | 0.00 | -1.82 | 4.00 | 0.50 | 1.96 | 1.94 | -0.22 | -0.14 | 3.97 | -0.69 | 3.45 | 0.00 | 0.00 | 1.05 |
| Models 9 | 3.88 | 5.36 | -0.29 | 2.00 | -0.36 | 1.47 | 2.72 | -0.11 | 0.08 | -1.44 | -0.11 | 0.78 | 1.41 | 0.00 | 1.10 |
| NLI 0 | 2.54 | 8.93 | -1.44 | 3.00 | 0.45 | 3.92 | 2.89 | 0.16 | -0.06 | 8.66 | 0.34 | 0.78 | 0.00 | -0.96 | 2.09 |
| NLI 1 | 2.32 | 10.71 | -2.97 | 15.00 | 0.41 | 2.21 | 2.64 | -0.38 | -0.46 | 9.03 | -1.26 | 1.10 | 1.41 | -0.96 | 2.77 |
| STS-B 1 | 2.48 | 8.93 | -1.63 | 11.00 | 0.26 | 4.41 | 3.07 | -0.11 | 0.11 | 9.39 | -0.34 | 1.88 | 0.00 | 0.00 | 2.82 |
| Multitask 0 | 5.23 | 10.71 | 0.29 | 3.00 | 0.09 | 3.43 | 3.53 | 0.90 | -0.92 | 11.19 | 0.80 | 3.76 | 0.00 | 0.00 | 3.00 |

Figure 6: Results of the off-the-shelf models/targets experiment. Rows correspond to off-the-shelf RoBERTa models obtained by downloading from HuggingFace model hub. Columns correspond to the General datasets group. Each value indicates intertraining gain w.r.t. using the PT model,

Target Dataset

| Source Dataset | Twitter Emoji | Twitter Emotion | Twitter Hate | Twitter Irony | Twitter Offensive | Twitter Sentiment | Twitter Stance abortion | Twitter Stance atheism | Twitter Stance climate | Twitter Stance feminist | Twitter Stance Hillary | avg |
|---|---|---|---|---|---|---|---|---|---|---|---|---|
| QA 3 | 7.71 | -11.12 | 4.65 | 1.66 | -2.33 | -8.15 | -4.29 | -4.55 | 0.59 | -0.35 | 2.03 | -1.28 |
| QA 5 | 0.26 | -2.11 | -1.52 | -1.40 | -1.98 | -0.10 | -5.36 | -5.91 | 0.00 | 3.86 | 5.76 | -0.77 |
| Summarization 9 | -1.73 | 0.00 | 0.98 | -1.79 | 1.16 | 1.95 | -1.79 | -3.64 | -0.59 | -1.05 | 3.73 | -0.25 |
| Classification 2 | -1.68 | 1.20 | 0.20 | -3.70 | 0.93 | 4.09 | 2.14 | 0.91 | -1.78 | -8.77 | 4.07 | -0.22 |
| Sentiment 1 | -1.68 | 1.20 | 0.20 | -3.70 | 0.93 | 4.09 | 2.14 | 0.91 | -1.78 | -8.77 | 4.07 | -0.22 |
| Summarization 0 | -1.16 | -0.14 | 1.25 | -3.19 | 0.93 | -1.58 | 1.07 | -1.82 | 0.00 | -1.40 | 5.42 | -0.06 |
| Classification 0 | -1.18 | -0.56 | 1.31 | 0.77 | 0.93 | 0.86 | 0.00 | -2.27 | -1.78 | -2.11 | 3.73 | -0.03 |
| Summarization 10 | -0.45 | 1.13 | 2.22 | -0.38 | 2.56 | 0.27 | 0.36 | -5.00 | 1.18 | 0.35 | -0.68 | 0.14 |
| Summarization 3 | -0.52 | -0.77 | 1.01 | 0.26 | 2.79 | -0.94 | 1.43 | -2.73 | -2.37 | -1.05 | 5.42 | 0.23 |
| Paraphrasing 1 | 8.46 | -0.21 | 1.25 | -4.85 | 2.09 | 0.39 | -0.71 | -5.45 | 1.78 | -1.05 | 2.03 | 0.34 |
| Summarization 7 | -1.02 | 0.77 | 1.78 | -1.28 | 2.91 | -1.55 | -1.79 | -3.18 | 0.59 | -3.16 | 9.83 | 0.36 |
| Summarization 5 | 7.99 | 0.00 | 1.55 | -2.30 | 1.40 | -0.95 | -1.07 | -1.36 | -1.18 | -1.40 | 1.69 | 0.40 |
| QA 1 | 9.36 | 0.28 | 0.34 | -4.59 | -2.33 | 1.34 | -3.93 | -5.91 | 4.73 | 1.40 | 4.07 | 0.43 |
| QA 2 | -0.87 | 0.63 | 1.62 | -1.66 | -1.40 | -0.07 | -0.36 | -2.73 | 4.14 | 1.75 | 5.76 | 0.62 |
| QG 0 | -0.87 | 0.63 | 1.62 | -1.66 | -1.40 | -0.07 | -0.36 | -2.73 | 4.14 | 1.75 | 5.76 | 0.62 |
| QA 6 | -1.23 | 0.42 | -0.27 | -1.28 | 1.40 | 1.22 | -0.71 | -4.09 | 1.78 | 3.16 | 9.49 | 0.90 |
| Summarization 6 | 8.58 | 1.06 | 1.62 | -3.32 | 0.58 | -1.07 | 1.07 | -2.27 | 0.59 | 1.75 | 1.69 | 0.94 |
| Summarization 2 | 9.93 | -0.28 | 1.45 | -0.89 | 0.12 | 0.35 | 1.07 | -3.64 | 2.37 | 0.35 | 2.03 | 1.17 |
| Summarization 4 | 9.58 | -0.35 | 3.54 | -0.77 | 0.93 | -1.27 | -1.43 | -1.36 | -2.37 | -0.70 | 9.15 | 1.36 |
| Summarization 1 | 9.94 | 0.77 | 3.00 | -0.77 | 1.28 | 2.04 | 0.00 | -3.64 | -1.78 | -1.05 | 6.78 | 1.51 |
| Classification 1 | 10.01 | 0.84 | -0.98 | -1.28 | 1.51 | 2.71 | -3.57 | -2.27 | 1.78 | 0.00 | 8.14 | 1.54 |
| Sentiment 0 | 10.01 | 0.84 | -0.98 | -1.28 | 1.51 | 2.71 | -3.57 | -2.27 | 1.78 | 0.00 | 8.14 | 1.54 |
| Summarization 8 | 9.58 | 1.34 | 2.56 | 1.40 | 0.81 | -0.16 | -1.43 | -4.55 | 1.18 | 2.81 | 6.10 | 1.79 |
| Paraphrasing 0 | 10.44 | -1.83 | 3.47 | -0.26 | -0.35 | 0.93 | -1.07 | -1.82 | 1.18 | 1.75 | 9.83 | 2.03 |
| QA 0 | 10.44 | -1.83 | 3.47 | -0.26 | -0.35 | 0.93 | -1.07 | -1.82 | 1.18 | 1.75 | 9.83 | 2.03 |
| QA 4 | 9.60 | -0.70 | 2.39 | -1.40 | 2.09 | 2.05 | 1.79 | -2.73 | 2.96 | 0.35 | 6.10 | 2.05 |

Figure 7: T5 off-the-shelf base models and targets. The intertrain gain over the pretrained model for each source (row) and target (column) datasets.

| | set | name | avg. gain across the General targets |
|---|---|---|---|
| 0 | QA_3 | allenai/t5-small-squad2-next-word-generator-squad | -1.28 |
| 1 | QA_5 | allenai/t5-small-squad11 | -0.77 |
| 2 | Summarization_9 | jazzisfuture/new_summary_t5_small | -0.25 |
| 3 | Classification_2 | mrm8488/t5-small-finetuned-imdb-sentiment | -0.22 |
| 4 | Sentiment_1 | mrm8488/t5-small-finetuned-imdb-sentiment | -0.22 |
| 5 | Summarization_0 | furyhawk/t5-small-finetuned-bbc-headline | -0.06 |
| 6 | Classification_0 | mrm8488/t5-small-finetuned-boolq | -0.03 |
| 7 | Summarization_10 | aseda/t5-small-finetuned-xsum | 0.14 |
| 8 | Summarization_3 | mengsay/t5-small-finetuned-gigaword | 0.23 |
| 9 | Paraphrasing_1 | hetpandya/t5-small-tapaco | 0.34 |
| 10 | Summarization_7 | bhuvaneswari/t5-small-text_summarization | 0.36 |
| 11 | Summarization_5 | bochaowei/t5-small-finetuned-cnn-wei1 | 0.40 |
| 12 | QA_1 | allenai/unifiedqa-t5-small | 0.43 |
| 13 | QA_2 | allenai/t5-small-squad2-question-generation | 0.62 |
| 14 | QG_0 | allenai/t5-small-squad2-question-generation | 0.62 |
| 15 | QA_6 | mrm8488/t5-small-finetuned-squadv2 | 0.90 |
| 16 | Summarization_6 | stevhliu/t5-small-finetuned-billsum-ca_test | 0.94 |
| 17 | Summarization_2 | furyhawk/t5-small-finetuned-bbc | 1.17 |
| 18 | Summarization_4 | bochaowei/t5-small-finetuned-cnn-wei0 | 1.36 |
| 19 | Summarization_1 | Frederick0291/t5-small-finetuned-billsum | 1.51 |
| 20 | Classification_1 | mrm8488/t5-small-finetuned-emotion | 1.54 |
| 21 | Sentiment_0 | mrm8488/t5-small-finetuned-emotion | 1.54 |
| 22 | Summarization_8 | airKlizz/t5-small-multi-combine-wiki-news | 1.79 |
| 23 | Paraphrasing/QA_0 | mrm8488/t5-small-finetuned-quora-for-paraphrasing | 2.03 |
| 24 | Paraphrasing/QA_4 | hetpandya/t5-small-quora | 2.05 |

Table 5: T5 models we used, collected from Hugging Face models hub. Models sorted by average gain over the General targets.

| Models | | @Top | Avg. | Max | number of datasets with $loss_n > 1$ |
|---|---|---|---|---|---|
| In-house | $loss_1$ | | 0.37 | 2.11 | 3/22 |
| | $loss_3$ | | 0.2 | 1.15 | 1/22 |
| Off-the-shelf | $loss_1$ | | 1.41 | 12.0 | 3/14 |
| | $loss_3$ | | 0.29 | 1.44 | 2/14 |

Table 6: Lost Gain per target is minimal when choosing the highest models, ranked by average intertraining gain on General datasets. Results are reported when selecting top rank model or best of 3 top rank models (@Top column). Columns represent the aggregation of the lost gain: average, max and the number of target datasets that lose at least one point. Rows represent two sets of experiments, in-house (with 22 models and 22 target datasets) or off-the-shelf (with 66 base modes and 14 target datasets).

of symmetry $s$ of the matrix M, considers the relations between $S$ and $V$, $s = (|S|-|V|)/(|S|+|V|)$, $s \in [-1, 1]$ if $s$ is close to -1 it means that $M$ is almost skew-symmetric (or anti-symmetric), if it is almost 1, it means that M is almost symmetric. If it around zero, it means that it neither symmetrical or skew-symmetric.

## H   U-shape

We analyse how the intertraining gains change when more target data is available. We find that while intertraining often improves results for small data size, the effect is decreasing with the size. Surprisingly, the decrease drops below zero and at some size increases again. This suggests an unexplained underlying behaviour, presumably of two competing effects, one that decreases gains with size and one that improves them (in general or towards 0). We produce three examples of the U-shape in Figures 8, 9, 10.

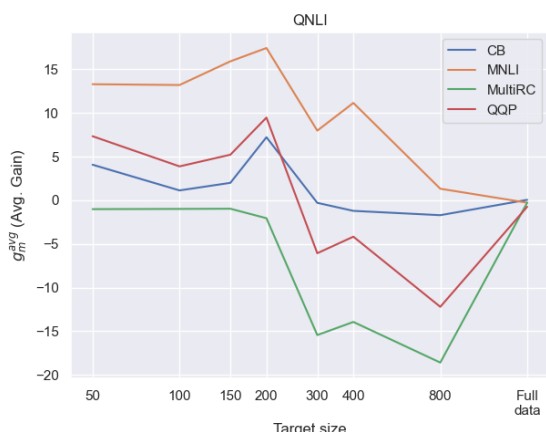

Figure 8: Gains of QNLI from intertraining with different amount of training data (X-axis) and different base models (lines).

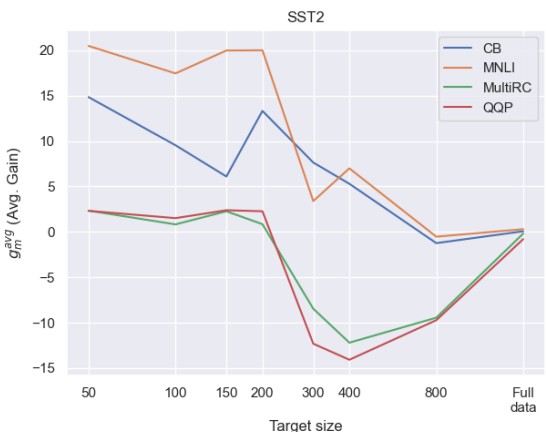

Figure 9: Gains of SST2 from intertraining with different amount of training data (X-axis) and different base models (lines).

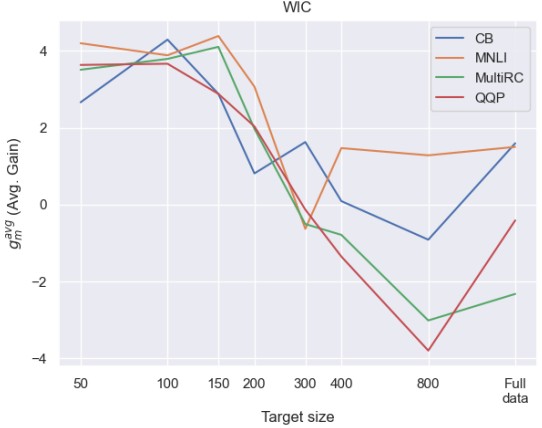

Figure 10: Gains of WIC from intertraining with different amount of training data (X-axis) and different base models (lines).

# I    Scores

We report the source and target scores of training on MNLI datasets with 20 seeds. Target scores are the average over the General datasets. In Fig. 11, we present the results. Evidently the two are not correlated.

## I.1    Forgetting

If PT's success comes from honing the parameters, shifting from them and *forgetting* the knowledge gained during pretraining is inadvisable in general (Chen et al., 2020) and possibly for intertraining (Pruksachatkun et al., 2020). With more training data, comes also more forgetting. This may also explain why most source models have a negative gain and intertraining hurts. Despite that, we observed in §6.1 that more source data empowers intertraining and improves gains. Following that observation, we analyze the importance of forgetting to the choice of a source model.

One common practice that causes forgetting is weight decay (Hanson and Pratt, 1988; Loshchilov and Hutter, 2019) – a regularization term added to the model updates. The decay term penalizes large weights, shrinking PT's large weights that are not necessary for the target objective.

For this experiment we use the following experimental setup: ADAMW (Loshchilov and Hutter, 2019) optimizer with weight decay 1 for decay and 0 (ADAM; Kingma and Ba, 2014) otherwise. L2 regularization is 0.1, results with other rates showed similar tendencies with effect size corresponding to the rate.

We find intermediate models trained with weight decay to be worse, but only if the pretraining did not include weight decay. Specifically, RoBERTa had weight decay during pretraining and T5 had not. We consider $g_{MNLI}^{avg}$ the average gain when source is MNLI and targets are General with and without weight decay. With RoBERTa as PT, the gain with decay was slightly better than without, by 0.02 points, while with T5 decay lost 3.3 points. These changes were not reflected on the source task performance.

Second, we limit the forgetting by adding a regularization forcing the model not to be far from the pretrained model. This can be seen as the complement of the previous method, rather than update to-

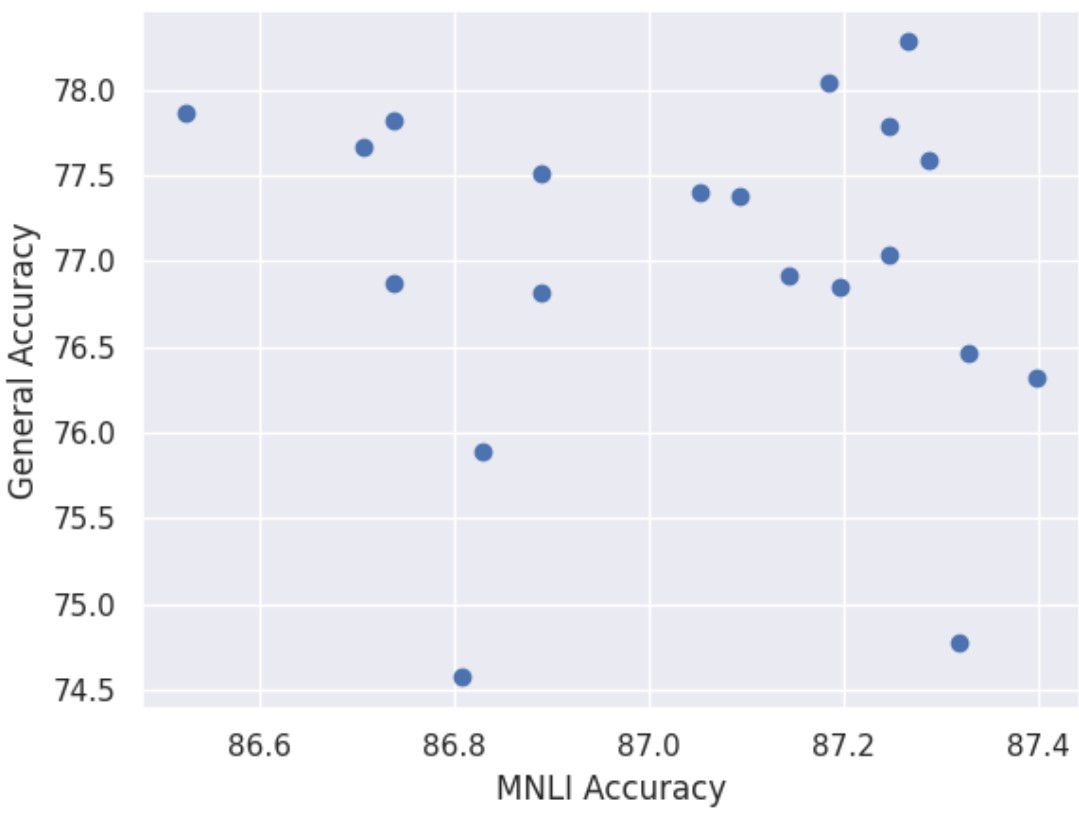

Figure 11: Source (MNLI) score against average score on General datasets after intertraining. Each point represents a different MNLI intermediate nodel trained on a different seed.

ward zero update toward the pretrained model. We find this method did not change much for MNLI (-0.28), but for models that hurt overall performance they prevented some performance loss, e.g. QQP (4.3). We note that while this regularization did not improve the top base model's gain, it did hurt the original finetuning on the source task (-5.2 points on MNLI). We further address the effect of source task score on base model quality in §6.2.

We also followed Kumar et al. (2022) method that should reduce forgetting with LP, but it had little effect.

All of the above findings imply that what determines a base model's effectiveness may be hidden in training hyperparameters. For example training on the same data and achieving similar results on the source, may still get quite different results on the target, depending on whether weight decay was used.

## J Architectures

Figs. 12, 13 depict the gains of In-house models trained on General datasets over the General datasets. In Fig. 7 we report the gains from training on off-the-shelf T5 models. Interestingly, QQP which is known as a bad source for RoBERTa, is among the best intermediate models for T5. Presumably, this is due to different training, where T5 generates the paraphrases rather than picks between several ones.

On a similar note, many of the top base models train on non-classification tasks, such as paraphrasing and question answering. This implies that the model weights converge to something quite general, learning linguistic traits that are not all discarded during finetuning. We say those are linguistic, in the sense that the language, and perhaps common knowledge are what makes the datasets similar, the tasks are quite different.

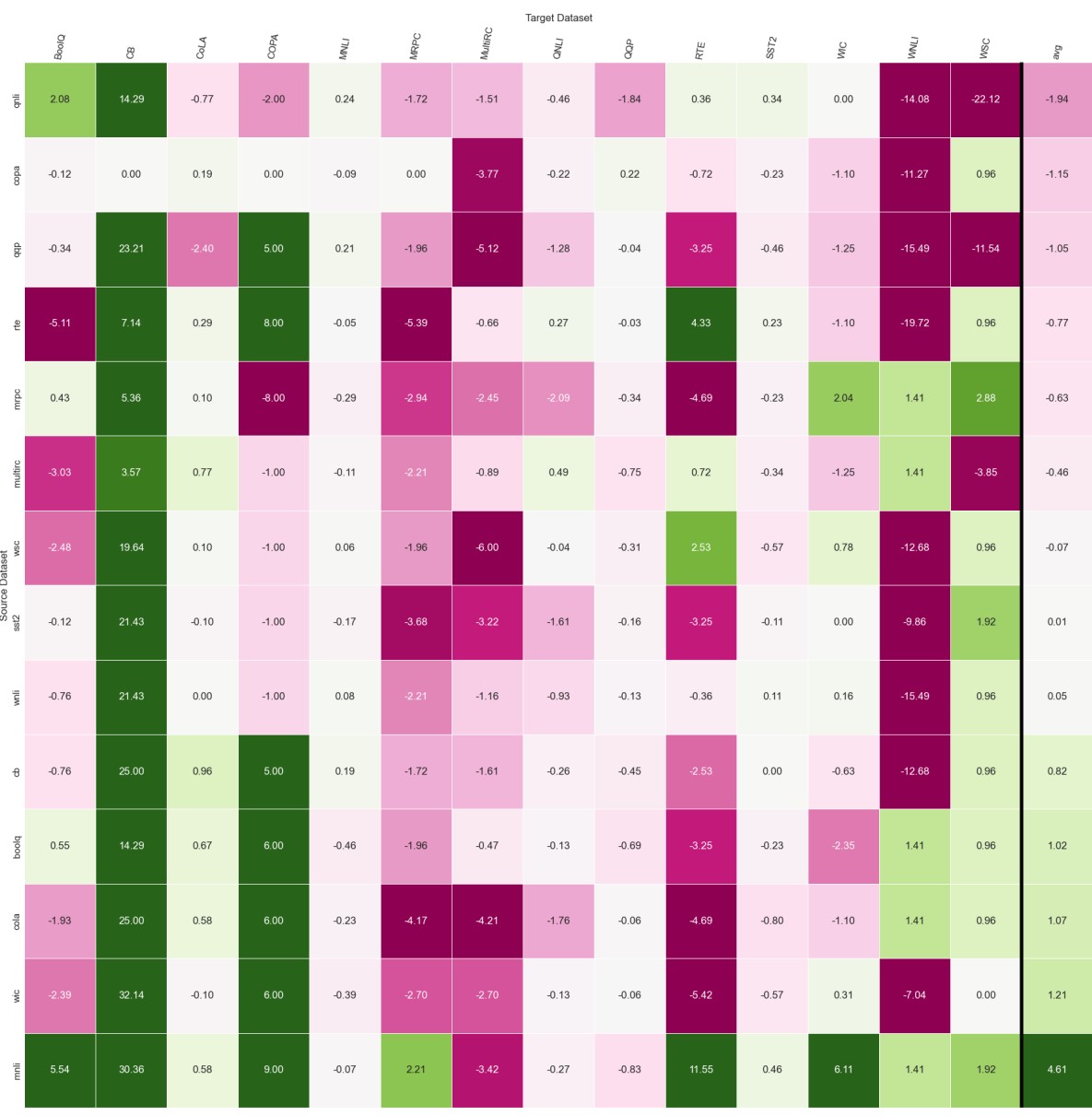

Figure 12: BERT General sources and targets. The intertrain gain over the pretrained model for each source (row) and target (column) datasets.

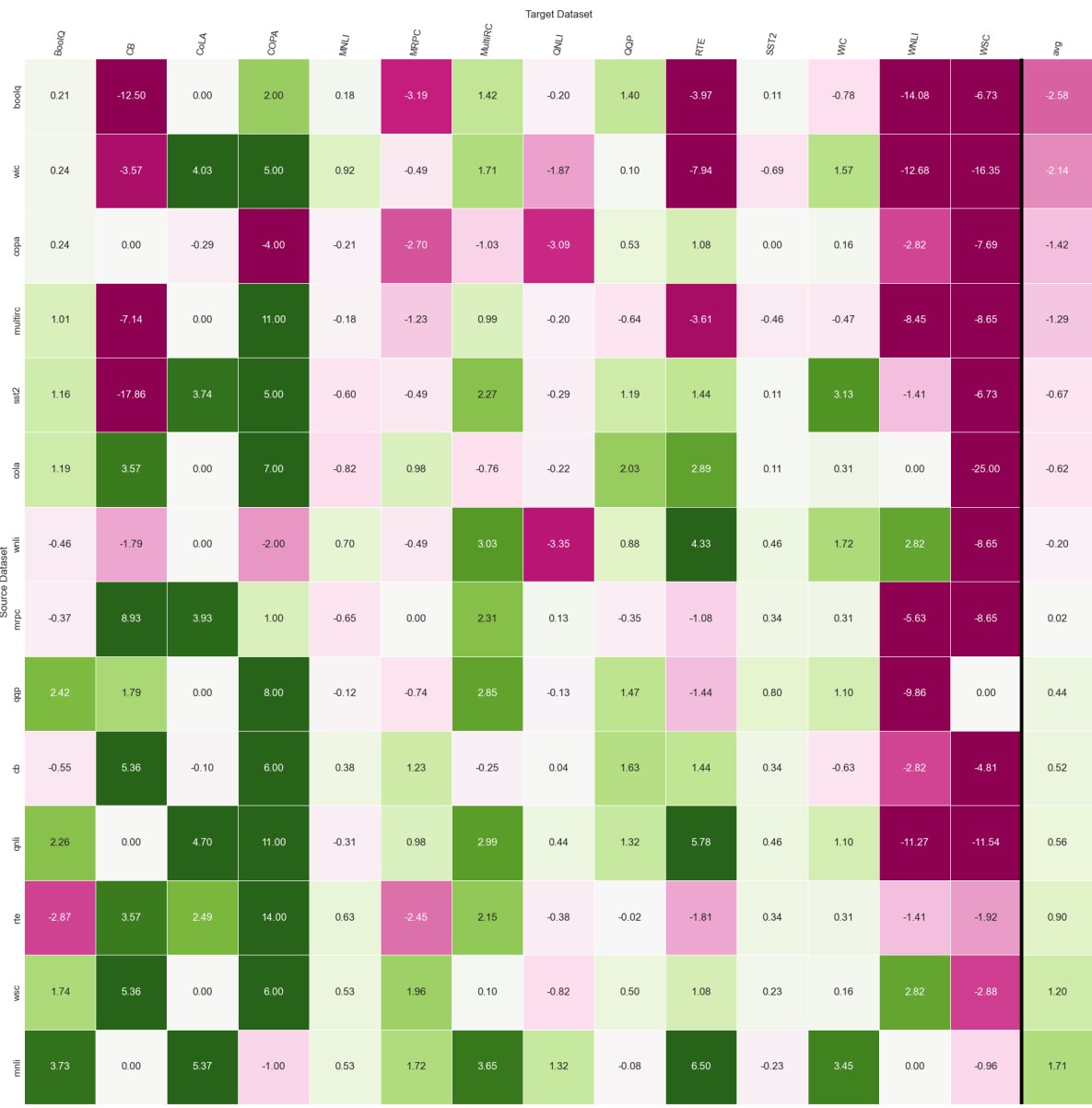

Figure 13: T5 General sources and targets. The intertrain gain over the pretrained model for each source (row) and target (column) datasets.