# OpenReview forum: "Where to start? Analyzing the potential value of intermediate models"
_EMNLP/2023/Conference — EMNLP 2023 Main_

### Official Review · Reviewer_mPxL · 2023-08-05

**Soundness:** 4

**Excitement:**

4: Strong: This paper deepens the understanding of some phenomenon or lowers the barriers to an existing research direction.

**Missing References:**

There is one related paper I can recall:
Exploring and Predicting Transferability across NLP Tasks, Vu et al 2020

**Paper Topic And Main Contributions:**

This work provides a comprehensive empirical study on intertraining, in which some source datasets might provide a better starting point for other datasets to start fine-tuning. The authors show that (i) the gain of intertraining can be analyzed independently of the target datasets. (2) Better intertraining models can be relatively easy to find by linear probing on MNLI. More discussions on the dataset size and model architecture are also provided.

**Reasons To Accept:**

1. This paper studies an interesting, and relatively unexplored problem.
2. Solid experiments are conducted on a wide range of datasets.
3. Clear takeaways, together with the model rankings, are provided for practitioners.

**Reasons To Reject:**

One potential concern about this work is we are gradually shifting away from specialized NLP models and the pretraining/finetuning paradigm. Both the model size and corpus size are scaled up hundreds of times nowadays, compared to single classification datasets and Roberta. I am not sure whether the conclusion in this paper still holds after scaling up. For example, can we find a single dataset like "MNLI" to rank models for more complex tasks/larger models?

**Reproducibility:**

3: Could reproduce the results with some difficulty. The settings of parameters are underspecified or subjectively determined; the training/evaluation data are not widely available.

**Reviewer Confidence:**

4: Quite sure. I tried to check the important points carefully. It's unlikely, though conceivable, that I missed something that should affect my ratings.

---

### Official Review · Reviewer_hzbj · 2023-08-05

**Soundness:** 4

**Excitement:**

3: Ambivalent: It has merits (e.g., it reports state-of-the-art results, the idea is nice), but there are key weaknesses (e.g., it describes incremental work), and it can significantly benefit from another round of revision. However, I won't object to accepting it if my co-reviewers champion it.

**Missing References:**

- Clifton Poth, Jonas Pfeiffer, Andreas Rücklé, and Iryna Gurevych. 2021. What to pre-train on? Efficient intermediate task selection. In Proceedings of the 2021 Conference on Empirical Methods in Natural Language Processing (EMNLP 2021).

- E. Bassignana, M. Mu ̈ller-Eberstein, M. Zhang, and B. Plank, “Evidence > intuition: Transferability estimation for encoder selection,” in EMNLP.

- Tu Vu, Tong Wang, Tsendsuren Munkhdalai, Alessandro Sordoni, Adam Trischler, Andrew Mattarella-Micke, Subhransu Maji, and Mohit Iyyer. Exploring and predicting transferability across NLP tasks. In Proceedings of the 2020 Conference on Empirical Methods in Natural Lan- guage Processing (EMNLP).

- Orion Weller, Kevin Seppi, and Matt Gardner. When to use multi-task learning vs intermediate fine-tuning for pre-trained encoder transfer learning. In Proceedings of the 60th Annual Meeting of the Association for Computational Linguistics (Volume 2: Short Papers), pp. 272–282, Dublin, Ireland, May 2022.

**Paper Topic And Main Contributions:**

The authors conduct an empirical analysis of intermediate training over 22 English classification tasks. They find that a model may exhibit strong performance even when the initial dataset that it was fine-tuned on is not semantically relevant to the target task. Based on their findings, they propose a ranking method for inferring such models in advance.

**Post rebuttal changes**

1. Increased Soundness score (+1)
2. Increased Excitement score (+1)

**Questions For The Authors:**

- Is Pearson Correlation a suitable metric? The weighted Kendall's tau coefficient seems to be more robust (see: LogME: Practical Assessment of Pre-trained Models for Transfer Learning)
- What is the computational efficiency of the proposed model ranking method (e.g. compared to brute-force fine-tuning?)

**Reasons To Accept:**

- The paper is well written and relatively easy to follow.
- The authors conducted extensive experiments.
- Open-sourcing the ranking method and model rankings will be valuable to the community.

**Reasons To Reject:**

- The authors do not use any standard baselines for comparing against the proposed method for ranking models (e.g. LogME: Practical Assessment of Pre-trained Models for Transfer Learning and MODEL SPIDER: Learning to Rank Pre-Trained Models Efficiently).
- The paper's overall contribution in relation to existing research remains ambiguous (see missing references for some examples). The authors' assertion that only a few works have explored intertraining/STILTs (lines 513-514) is inaccurate.

**Reproducibility:**

3: Could reproduce the results with some difficulty. The settings of parameters are underspecified or subjectively determined; the training/evaluation data are not widely available.

**Reviewer Confidence:**

4: Quite sure. I tried to check the important points carefully. It's unlikely, though conceivable, that I missed something that should affect my ratings.

---

### Official Review · Reviewer_seM1 · 2023-08-10

**Soundness:** 4

**Ethical Concerns:**

Yes

**Excitement:**

4: Strong: This paper deepens the understanding of some phenomenon or lowers the barriers to an existing research direction.

**Paper Topic And Main Contributions:**

Existing works find a fact that fine-tuned models on targer domain may provide a better starting point for a new finetuning process. Such a phenomenon dubed inter-training schema.
In this paper, author perform a systematic analysis of such scheme over a wide range of  tasks and get insight from these empirical studies. The research is begin with two observations:

(1) some target datasets are intertraining-sensitive;

(2) some base models are of high quality, finetuning on them provides consistent improvements on target datasets.

Based on aforementationed observations,     a preferable base model can be selected independently of the target dataset. They substantiates their observation of independence by conducting experiments on a comprehensive set of target datasets and base models.

In addition to above findings, they analyze attributes of the source and target datasets that affect gains.
They propose a practical approach to efficiently select models in a real-world setting, and make best models currently found  publicly available.

**Questions For The Authors:**

see above.

**Reasons To Accept:**

1.  This paper presents an empirical evaluation of the inter-training scheme. The authors propose several viewpoints that have been overlooked in previous studies and validate them through experiments. These viewpoints provide valuable guidance for future research work.
2. The authors provide a comprehensive discussion on the potential impacts of the model, source dataset, and target dataset, which is highly convincing.
3.  The paper is well-written.

**Reasons To Reject:**

1. There are some formatting issues in this paper, such as the tables, which the authors should carefully address. However, these formatting issues do not affect the technical contributions of the paper.

2. The experiments in this paper mainly focus on common datasets like GLUE benchmark, NLI, and Twitter. Including a more diverse range of test datasets would enhance the persuasiveness of this work. The authors could also consider whether the proposed method can be generalized to the visual or multimodal domains.

3. Adding a figure for regression  in Section 5 would provide better clarity and aid in understanding.

**Reproducibility:**

4: Could mostly reproduce the results, but there may be some variation because of sample variance or minor variations in their interpretation of the protocol or method.

**Reviewer Confidence:**

4: Quite sure. I tried to check the important points carefully. It's unlikely, though conceivable, that I missed something that should affect my ratings.

---

### Official Review · Reviewer_H4LS · 2023-08-10

**Soundness:** 4

**Excitement:**

4: Strong: This paper deepens the understanding of some phenomenon or lowers the barriers to an existing research direction.

**Paper Topic And Main Contributions:**

This paper empirically examines underlying factors that dominate the performance gain of "intertraining", where the pre-trained models are trained on a source dataset first before further fine-tuning on a target downstream dataset. The authors conduct extensive experiments and present several interesting findings: the contributions to the performance gain of the source dataset and the target dataset are nearly independent; the target dataset sensitivity could be efficiently estimated by linear probing; the source and target dataset sizes play an important role in performance gain. I think this work is pretty impressive as it provides a principled guideline for fine-tuning model selection via comprehensive experiments and studies. But I also find that there are some issues in the main content: the authors rank models by their averaged performance gain, which I believe could be problematic as the mean conceal extreme values in the distribution and is less informative in some comparisons, e.g., base models from MultiRC seem to results fewer extremely low results (<-5) than those from QQP, yet base models from QQP have a larger averaged performance gain and thus rank higher than those from MultiRC. In Section 5, the authors adopt a linear regression model to justify the independency between the source and target datasets. The authors report an MSE of 4.2, but I have no idea whether this error is acceptable for the independency to hold. I think the authors need to clarify this part, e.g., under this MSE, how much performance gain (in percentage) could be attained. In the end, the writing of this paper could be improved, as I see some typos in the main text.

**Reasons To Accept:**

1. Thorough experiments and interesting findings.

**Reasons To Reject:**

See my comment above.

**Reproducibility:**

3: Could reproduce the results with some difficulty. The settings of parameters are underspecified or subjectively determined; the training/evaluation data are not widely available.

**Reviewer Confidence:**

2: Willing to defend my evaluation, but it is fairly likely that I missed some details, didn't understand some central points, or can't be sure about the novelty of the work.

**Typos Grammar Style And Presentation Improvements:**

Line 94: ’source’ -> ‘source’
Line 109: max should not be in italics

---

### Meta-Review · Area_Chair_SRLs · 2023-09-19

**Recommendation:** 5

**Metareview:**

This paper studies the value of an intermediate model in an inter-training setting, which corresponds to a certain style of transfer learning, where the pretrained language model is being trained on different tasks before the task of interest. Experimental results on a wide variety of benchmarks show the importance of the size of target and source datasets, as well as the value of the base model. The paper ranks different models for tasks by the average performance gain under this inter-training setting.

Overall, the reviewers seem compelled by the detailed empiricism of the work and present suggestions for future research such as extensions to other domains or replication in a scaled up setting. Some issues pointed out regarding interpretability of MSE scores and aggregation of scores seem to be addressed by the authors’ response.

We hope the authors include the suggested changes and clarifications in the next iteration of the paper.

---

### Decision · Program_Chairs · 2023-10-07

**Decision:**

Accept-Main

**Comment:**

This paper studies the value of an intermediate model in an inter-training setting, which corresponds to a certain style of transfer learning, where the pretrained language model is being trained on different tasks before the task of interest. Experimental results on a wide variety of benchmarks show the importance of the size of target and source datasets, as well as the value of the base model. The paper ranks different models for tasks by the average performance gain under this inter-training setting.

Overall, the reviewers seem compelled by the detailed empiricism of the work and present suggestions for future research such as extensions to other domains or replication in a scaled up setting. Some issues pointed out regarding interpretability of MSE scores and aggregation of scores seem to be addressed by the authors’ response.

We hope the authors include the suggested changes and clarifications in the next iteration of the paper.